# PERMUTE-AND-FLIP: AN OPTIMALLY STABLE AND WATERMARKABLE DECODER FOR LLMS

**Xuandong Zhao**
UC Berkeley
xuandongzhao@berkeley.edu

**Lei Li**
Carnegie Mellon University
leili@cs.cmu.edu

**Yu-Xiang Wang**
UC San Diego
yuxiangw@ucsd.edu

## ABSTRACT

In this paper, we propose a new decoding method called Permute-and-Flip (PF) decoder. It enjoys stability properties similar to the standard sampling decoder, but is provably up to 2x better in its quality-stability tradeoff than sampling and never worse than any other decoder. We also design a cryptographic watermarking scheme analogous to Aaronson (2023)'s Gumbel watermark, but naturally tailored for PF decoder. The watermarking scheme does not change the distribution to sample, while allowing arbitrarily low false positive rate and high recall whenever the generated text has high entropy. Our experiments show that the PF decoder (and its watermarked counterpart) significantly outperform(s) naive sampling (and its Gumbel watermarked counterpart) in terms of perplexity, while retaining the same stability (and detectability), hence making it a promising new approach for LLM decoding. The code is available at https://github.com/XuandongZhao/pf-decoding.

## 1 INTRODUCTION

Large language models (LLMs) (OpenAI, 2022; 2023b; Bai et al., 2022; Touvron et al., 2023) have become increasingly popular in recent years due to their ability to generate human-like text and solve many tasks through a natural chatbot interface.

A language model predicts the next word in a sentence using a real-value function $u(\cdot; \text{prompt}, \text{prefix}) : \mathcal{V} \to \mathbb{R}$, known as *logits*, which encodes the model's preferences on which word to choose. Here $\mathcal{V}$ is the vocabulary space (typically a large discrete set of words); the "prompt" describes the task of interest; and "prefix" includes all preceding words that have been generated so far.

A language model *decoder* refers to a possibly randomized function that takes a prompt text $x$, API access to the *logits* function as input, and outputs a sentence $y_{1:n}$.

**The main thrust of this paper** is to introduce a new decoder, termed *Permute-and-Flip decoding*, work out some of its intriguing properties with an application to watermarking LLM text, and hopefully convince readers that it deserves a shot at your next LLM application.

## 2 PROBLEM SETUP AND SUMMARY OF RESULTS

Before diving in, let's set up the stage with a quick tour of existing decoding methods and have a brief look into how a language model decoder can be evaluated.

**Popular existing decoding methods** fall into three categories: (1) Planning-based methods such as beam search that aims at maximizing the sequence likelihood; (2) sampling-based methods that recursively sample from the next-word distribution, e.g., the soft(arg)max transform of the logits

$$\text{Softmax sampling: } y_t \sim p(y) = \frac{e^{u(y|x,y_{1:t-1})/T}}{\sum_{\tilde{y}} e^{u(\tilde{y}|x,y_{1:t-1})/T}} \quad (1)$$

where $T$ is the *temperature* parameter; and (3) greedy methods such as greedy decoding that simply outputs $y_t = \arg\max_{y \in \mathcal{V}} u(y|x, y_{1:t-1})$ as well as its Top $p$ (Holtzman et al., 2020) and Top $k$ sampling variants that interpolate greedy and sampling methods.

**Performance metrics.** How do we compare different decoding methods? More generally, how do we evaluate LLM-generated text? These are questions far from being settled. Naturally, if there is a (possibly task-dependent) performance metric $U_x : \mathcal{V}^n \to \mathbb{R}$ one can define, then the optimal decoder would be the one that outputs $y_{1:n}^* = \arg\max_{y_{1:n} \in \mathcal{V}^n} U_x(y_{1:n})$. Often $U_x$ is instantiated to be the sequence likelihood $\sum_{t=1}^n \log p(y_t|x, y_{1:t-1})$ which is equal to $\sum_{t=1}^n u_t(y_t)$.

Recent works (Ippolito et al., 2019; Wiher et al., 2022), however, report that strategies that aim at maximizing sequence likelihood often result in texts that are more repetitive and less effective in some downstream tasks than those from the sampling-based methods (Holtzman et al., 2020). Depending on what the task is, there is not a one-size-fits-all performance metric, therefore is no single decoding method that works well for all tasks.

For the moment, let us stash the disputes on how to best evaluate an LLM-generated text and focus on designing methods that maximize any user-specified utility function. In fact, we will also give up on solving the sequence-level utility maximization problem[1] and simply maximize a *per-step* utility function $u_t : \mathcal{V} \to \mathbb{R}$.

$u_t$ can simply be the logits function that LLMs output, which may have already accounted for potential future utility (like the $Q$ function in reinforcement learning) since the transformer-based language model had access to future texts during pre-training. Or $u_t$ can be explicitly augmented with structure-inducing regularizers such as a lookahead heuristic as in A* decoding (Lu et al., 2022), a retrieval-based term for fact-checking (Lewis et al., 2020), or an entropy bonus for promoting diversity (Meister et al., 2020).

**Our goal** is thus to construct a possibly randomized algorithm $\mathcal{A}$ that takes $u_t$ as an input and outputs $y_t \in \mathcal{V}$ that aims at maximizing $\mathbb{E}_{y_t \sim \mathcal{A}_{u_t}}[u_t(y_t)]$ as much as possible. In the remainder of the paper, we will simply take $u_t$ as "logits" for a concrete exposition — all results are valid when $u_t$ is instantiated otherwise.

**Other constraints / consideration.** Why doesn't the trivial greedy decoder work? That's because there are other considerations besides text quality when selecting LLM decoders. For example, **computational efficiency and latency** are hugely important, since each API call to the *logits* function is costly. The **diversity** of the generated text is also important, especially for creative tasks.

Moreover, the decoding procedure should be **watermarkable** (Aaronson, 2023; Kirchenbauer et al., 2023; Zhao et al., 2023; Kuditipudi et al., 2024) in the sense that one should be able to inject subtle statistical signals that can be retrieved when given a secret key, to *prove* that the text is generated by this particular language model. Being watermarkable prevents the LLM from being used for malicious purposes such as scams (Weidinger et al., 2021), fake news (Zellers et al., 2019), and plagiarism (Stokel-Walker, 2022).

In addition to the above, one may also hope the decoding algorithm to be **stable against small perturbations** to the *logits*. Specifically,

**Definition 2.1** (Stability). We say a decoding algorithm $\mathcal{A}$ is $L$-stable if for any prompt $x$, prefix $y_{\leq t}$, and for any perturbed $\tilde{u}$ such that $\|\tilde{u} - u\|_\infty \leq \delta$, the log-probability ratio satisfies

$$\left| \log \left\{ \frac{p_{\mathcal{A}(\tilde{u}(\cdot|x, y_{\leq t}))}(y)}{p_{\mathcal{A}(u(\cdot|x, y_{\leq t}))}(y)} \right\} \right| \leq L\delta \quad \forall y \in \mathcal{V}.$$

The stability helps to avoid catastrophic failure in the scenarios where the logits may be subject to data poisoning (Zhang et al., 2022; Lin et al., 2021) or jailbreaking attacks (Zhang et al., 2023; Zhao et al., 2024b). Furthermore, *stability* implies an intuitive notion of *diversity*, suggesting that tokens with similar logits should have comparable probabilities of being selected. For further discussion and examples, please refer to Appendix B.

Inspecting the decoding methods along the aforementioned dimensions, we notice that planning-based methods fail to be computationally efficient. While greedy decoding is efficient and has relatively low perplexity, its generated texts are neither diverse nor watermarkable (at least not using existing techniques). The sampling-based methods, however, are both watermarkable and diverse. In addition, softmax sampling is known to be 2-stable, while the others that we have discussed are not stable.

---

[1]It is known to be NP-Complete (Chen et al., 2018).

Table 1: Comparison of different decoding methods against five desiderata.

| Methods | Perplexity | Computational Efficiency | Diversity | Watermark | Stability |
|---|---|---|---|---|---|
| Search (e.g., Beam) | Lowest | ✗ | ✗ | ✗ | ✗ |
| Greedy | Low | ✓ | ✗ | ✗ | ✗ |
| Softmax Sampling | Moderate | ✓ | ✓ | ✓ | ✓ |
| Top-$p$ Sampling | Low (for small $p$) | ✓ | Depends on $p$ | ✓ | ✗ |
| Top-$k$ Sampling | Low (for small $k$) | ✓ | Depends on $k$ | ✓ | ✗ |
| PF Sampling (ours) | Lower than Softmax | ✓ | ✓ | ✓ | ✓ |

**Fact 2.2.** *Softmax sampling decoding using* (1) *with temperature $T$ satisfies* $(2/T)$-*stability.*

*Proof.* The result is implied by the differential privacy guarantee of exponential mechanism (McSherry & Talwar, 2007, Theorem 6). □

The pros and cons of different decoding methods are summarized in Table 1. From the table, we can see that there is a clear tradeoff between minimizing perplexity and preserving other properties. In particular, softmax sampling is the only method that checks all boxes, and the only one that is stable among existing decoders. This observation begs the following research question: *Is there a decoding method that is as stable as softmax sampling, but has lower perplexity?*

In this paper, we answer this question affirmatively by bringing in a technique called Permute-and-Flip sampling. Our contributions are fourfold.

1. We introduce Permute-and-Flip decoding — a new decoding algorithm for language models based on recent development in a very different context (McKenna & Sheldon, 2020).
2. We demonstrate that existing results from McKenna & Sheldon (2020) already imply that:
   - Permute-and-Flip decoding is provably stable.
   - The stability-perplexity tradeoff of the PF decoding is Pareto-optimal. In particular, when compared to softmax sampling, PF decoding has up to 2x smaller expected suboptimality while having the same stability parameter $L$.
3. We designed an analog of Aaronson (2023)'s Gumbel-Watermark for PF decoder, called the PF watermark. We show that the watermarked PF decoder samples from a distribution that is computationally indistinguishable from the non-watermarked PF decoder, and the detection procedure has precisely controlled false positive rate (FPR) and high power in identifying watermarked texts.
4. We empirically demonstrate that on open-generation tasks, PF watermark achieves the best balance of the highest detection accuracy and lowest perplexity compared to the baselines.

Overall, our proposed permute-and-flip decoding method provides a promising approach to balancing the tradeoff between perplexity and stability in LLM decoding while also admitting watermarking capabilities.

**Related work and novelty.** PF sampling was invented in the differential privacy (DP) literature (McKenna & Sheldon, 2020). Its stability properties are well-understood. The stability of Softmax sampling is also well-known (McSherry & Talwar, 2007). Our contribution is in applying this method to LLM decoding and connecting these known theoretical results to the broader ML audience. To our knowledge, the PF watermark is new to this paper. The design of the watermark leverages the Report-Noisy-Max interpretation of the PF sampling (Ding et al., 2021) which allows a similar pseudo-random function like the work of Aaronson (2023) to be applied. A more thorough discussion of the related work is given in Appendix A.

## 3 PERMUTE-AND-FLIP DECODING ITS PROPERTIES

The Permute-and-Flip decoding iteratively generates the next token by a simple procedure that uses only the logits. It involves first randomly permuting the vocabulary, then flipping sequence of biased coins according to the permuted sequence until the first "head" is seen (see Algorithm 1).

---

**Algorithm 1** Permute and Flip (PF) Decoding

---
1: **Input:** prompt $x$, language model $\mathcal{M}$, temperature $T$.
2: **for** $t = 1, 2, \cdots$ **do**
3:   Logits $u_t \leftarrow \mathcal{M}([x, y_{1:t-1}])$.
4:   Find $u_t^* \leftarrow \max_{y \in \mathcal{V}} u_t(y)$.
5:   Permute : Shuffle the vocabulary $\mathcal{V}$ into $\tilde{\mathcal{V}}$.
6:   **for** $y \in \tilde{\mathcal{V}}$ **do**
7:     Flip : Draw $Z \sim \text{Bernoulli}\left(\exp\left(\frac{u_t(y) - u_t^*}{T}\right)\right)$.
8:     **if** $Z = 1$, **then** assign $y_t \leftarrow y$ and **break**.
9:   **end for**
10: **end for**
11: **Output:** Generated sequence $y = [y_1, ..., y_n]$.

---

Permute-and-flip makes words with higher logits exponentially more likely — even more so than Softmax sampling (Eq. 1). To see this, one may consider a rejection sampling algorithm for obtaining a sample from Eq. (1), which repeats the following procedures until it halts.

1. Uniformly samples $y \in \mathcal{V}$,

2. Return it with probability:

$$p(y)/p(y^*) = \exp\left((u_t(y) - u_t(y^*))/T\right).$$

This procedure differs from PF sampling in that it samples $y$ *with replacement*, whereas PF sampling samples $y$ *without replacement*, giving PF sampling a higher likelihood of producing $y^*$.

PF sampling was initially proposed in McKenna & Sheldon (2020) as a differentially private selection mechanism that has better utility than the more well-known exponential mechanism (McSherry & Talwar, 2007). McKenna & Sheldon (2020) also derived a plethora of theoretical properties of the PF sampling. The following theorem summarizes these results in the language of LLM decoding.

**Theorem 3.1.** *Let the logits function be $u$ and $u^* = \max_{y \in \mathcal{V}} u(y)$. Let $\text{PF}(u)$ be the distribution of PF-sampling, and $\text{Softmax}(u)$ be the distribution in (1), both with temperature parameter $T$. The following statements are true.*

*1. (**Same stability**) PF-Sampling is $(2/T)$-stable.*

*2. (**Nearly greedy**) PF-sampling obeys,*

$$\mathbb{E}_{y \sim \text{PF}(u)}[u(y)] \geq u^* - T \log |\mathcal{V}|.$$

*3. (**"Never worse"**) For the same $T$, PF-sampling is never worse than Softmax-sampling.*

$$\mathbb{E}_{y \sim \text{PF}(u)}[u(y)] \geq \mathbb{E}_{y \sim \text{Softmax}(u)}[u(y)]$$

*4. (**"Up to 2x better"**) There exists logits $u$ such that PF-sampling is 2x smaller in terms of suboptimality,*

$$\mathbb{E}_{y \sim \text{PF}(u)}[u^* - u(y)] \leq \frac{1}{2}\mathbb{E}_{y \sim \text{Softmax}(u)}[u^* - u(y)].$$

*5. (**Optimal stability-perplexity tradeoff**) For any decoder $P$ that is $2/T$-stable, if there exists $u$ such that*

$$\mathbb{E}_{y \sim P(u)}[u(y)] > \mathbb{E}_{y \sim \text{PF}(u)}[u(y)]$$

*then there must be another $\tilde{u}$ such that*

$$\mathbb{E}_{y \sim P(\tilde{u})}[\tilde{u}(y)] < \mathbb{E}_{y \sim \text{PF}(\tilde{u})}[\tilde{u}(y)].$$

*Proof.* The theorem follows directly from McKenna & Sheldon (2020), specifically Theorem 1, Corollary 1, Theorem 2, Proposition 4, and Proposition 6. □

The first statement shows that the PF decoder enjoys exactly the same stability parameter as in Fact 2.2. The second statement provides a worst-case bound on how far PF-sampling is away from greedy-decoding as a function of the temperature $T$ in terms of the likelihood achieved. The third and fourth statements show that PF-sampling is always "more greedy" than softmax-sampling. The last statement shows that PF-sampling is not dominated by any other decoder that is equally stable (as in Definition 2.1), thus *Pareto optimal*. These results provide strong justification on

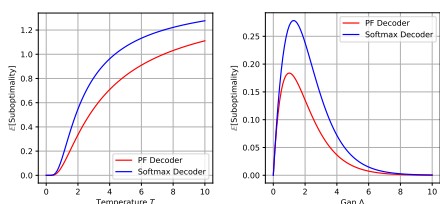

Figure 1: Comparing PF decoder vs Softmax decoder using Example 3.2.

the superiority of the permute-and-flip decoder over the standard softmax sampling in minimizing perplexity.

Let's consider a simple example to compare PF decoder and Softmax decoder.

**Example 3.2.** Let the $|\mathcal{V}| = 2$ and the corresponding logits be $[\Delta, 0]$ for gap $\Delta > 0$. Softmax decoder chooses the suboptimal token with probability $1/(1 + e^{\Delta/T})$, while PF decoder chooses it w.p. $1/(2e^{\Delta/T})$.

Since $1/(1 + x) > 1/(2x)$ for all $x > 1$, the probability that the suboptimal token is chosen in PF sampling is strictly smaller than that of Softmax sampling. As shown in Figure 1, on the left, we fix the Gap $\Delta = 3.0$ and vary the temperature $T$. On the right, we fix $T = 1.0$ and vary $\Delta$. PF beats Softmax in all cases.

## 4 REPORT-NOISY-MAX AND WATERMARKING

Next, we turn to the well-motivated problem of watermarking LLM generated text. The watermarking problem aims at embedding a secret message in the generated text that (essentially) reads "Beware! I am written by an AI!".

The hope is that this message can be seen by anyone who has access to a *secret key*, while ensuring that the watermarked version of the LLM generates text that has *almost the same* distribution as (or at least very similar) to the original LLM.

More formally, a watermarking scheme includes a "Watermark" function that injects the watermark and a "Detect" function that takes a suspect text sequence $y_{1:n}$ as input and outputs a prediction of 1 ("It is watermarked!") or 0 ("It is not!").

A wrong accusation of non-watermarked text as watermarked is called a *false positive*. A failure to detect a watermarked text is called a *false negative*. The performance of a watermark is measured by its detection *power* (i.e., $1-$*false negative rate*) at a given *false positive rate*.

There are many other necessary properties for a watermarking scheme to be useful, such as *low-overhead*, *model-agnostic* detection, and *resilience to edits* and other evasion attacks. We refer readers to the slide deck of Aaronson (2023) and the related work section of (Zhao et al., 2024a) for a review of these desiderata and known results.

Among the recent attempts, two popular watermarking schemes perform satisfactorily on all the above criteria.

**Gumbel Watermark (Aaronson, 2023)** that uses a "traceable" pseudo-random softmax sampling when generating the next word.

**Green-Red Watermark (Kirchenbauer et al., 2023)** that randomly splits the vocabulary into Green and Red then slightly increases the logits for green tokens.

Both of them determine their pseudo-random seeds chosen according to the $m$ preceding tokens of the current token being generated. We will focus on explaining the Gumbel watermark as it is more closely related to our approach.

**Aaronson (2023)'s Gumbel watermark.** The key idea of the Gumbel watermark leverages the "Gumbel-Max Trick", which states that:

**Fact 4.1** (Gumbel, 1948). *The softmax sampling in* (1) *is equivalent to the following procedure*

$$y_t = \arg\max_{y \in \mathcal{V}} \frac{u_t(y)}{T} + G_t(y) \tag{2}$$

*where* $G_t(y) \sim$ *Gumbel*$(0, 1)$ *i.i.d for each* $t, y$.

Gumbel noise can be generated using a uniform r.v.

$$\text{Gumbel}(0, 1) \sim -\log\left(\log(1/\text{Uniform}([0, 1]))\right).$$

So given a random vector $r_t \sim (\text{Uniform}([0, 1]))^{|\mathcal{V}|}$, we can write $G_t(y) = -\log(-\log(r_t(y)))$.

The Watermark stage for the Gumbel-watermark essentially replaces $\text{Uniform}([0, 1])$ with a *pseudo-random function* $r_t(y) = F_{y_{t-m:t-1}, \mathsf{k}}(y)$. Given the secret key $\mathsf{k}$, the *pseudo-random function* is a deterministic function with range $[0, 1]^{\mathcal{V}}$, but over the distribution of the secret key $\mathsf{k}$, $r_t$ is computationally indistinguishable from sampled from truly i.i.d. uniform distribution, which ensures that the distribution of $y_t$ in the watermarked model is computationally indistinguishable to the unwatermarked distribution (1).

At Detect phase of the the Gumbel watermark, the auditor who has access to the key $\mathsf{k}$ may compute

$$\text{TestScore}_{\text{Gumbel}}(y_{1:n}) = \sum_{t=m+1}^{n} -\log(1 - r_t(y_t)).$$

If $y_{1:n}$ is *not* generated from the watermarked model, then the test statistic is a sum of exponential random variable thus $\mathbb{E}[\text{TestScore}(y_{1:n})] = n - m$. Meanwhile, if $y_{1:n}$ is generated by the Gumbel watermarked model,

$$\mathbb{E}[\text{TestScore}(y_{1:n})] = \sum_{t=m+1}^{n} \mathbb{E}\left[\sum_{y \in \mathcal{V}} p_t(y) H_{\frac{1}{p_t(y)}}\right] \tag{3}$$

$$\geq (n - m) + \left(\frac{\pi^2}{6} - 1\right) \sum_{t=m+1}^{n} \mathbb{E}\left[\text{Entropy}[p_t(\cdot)]\right]. \tag{4}$$

where $p_t := \text{Softmax}(u_t/T)$, $H_\alpha := \int_0^1 \frac{1-x^\alpha}{1-x} dx$ is Euler's Harmonic number and Entropy denotes the standard Shannon entropy (in nats) for a discrete distribution, i.e., $\text{Entropy}[p] = -\sum_{y \in \mathcal{V}} p(y) \log p(y)$. The above results were described in Aaronson (2023)'s talk, but without detailed proofs. To be self-contained, we added formal statements and proofs about the Gumbel watermarks in Appendix D.3 (see Theorem D.3 and D.6).

**Permute-and-Flip as ReportNoisyMax.** It turns out that the Permute-and-Flip sampling has a similar equivalent Report-Noisy-Max form. Instead of Gumbel noise, it is the exponential noise that are added to the logits. This less-known fact is due to Ding et al. (2021)

**Fact 4.2** (Ding et al., 2021, Theorem 5). *Permute-and-Flip Sampling in Algorithm 1 with parameter* $T$ *is equivalent to*

$$y_t = \arg\max_{y \in \mathcal{V}} \frac{u_t(y)}{T} + E_t(y). \tag{5}$$

*where* $E_t(y) \sim$ *Exponential*$(1)$ *i.i.d. for each* $t, y$.

Leveraging this fact, in the remainder of the section, we develop a watermarking scheme for Report-NoisyMax that is analogous to the Gumbel-watermark.

**Permute-and-Flip watermark.** The natural idea is to replace the exponential noise $E_t(y)$ with a pseudo-random version that depends on a secret key and a prefix with length $m$. Observe that $\text{Exponential}(1) \sim -\log(\text{Uniform}([0, 1]))$, thus the standard pseudo-random function that generates uniform random variables can be used. In the detection phase, we compute:

$$\text{TestScore}_{\text{PF}}(y_{1:n}) = \sum_{t=m+1}^{n} -\log(r_t(y_t)).$$

Note that this is a simple change of sign of $r_t(y_t)$ comparing to the test score of the Gumbel watermark. Detailed pseudo-code for how the watermark works are given in Algorithm 2 and Algorithm 3.

---

**Algorithm 2** PF watermarking: Watermark

---

1: **Preparation:** Randomly sample a watermark key k
2: **Input:** Prompt $x$, language model $\mathcal{M}$, pseudo-random function $F$, watermark key k, temperature $T$
3: **for** $t = 1, 2, \cdots$ **do**
4:     Compute logits: $u_t \leftarrow \mathcal{M}([x, y_{1:t-1}])$
5:     Generate a pseudo-random vector $r_t(\cdot)$ using $r_t(y) := F_{y_{t-m:t-1}, \mathsf{k}}(y)$ for $y \in \mathcal{V}$
6:     Select the next token $y_t$ using

$$y_t = \arg\max_{y \in \mathcal{V}} \left( \frac{u_t(y)}{T} - \log r_t(y) \right) \tag{6}$$

7: **end for**
8: **Output:** Watermarked sequence $y = [y_1, ..., y_n]$

---

**Algorithm 3** PF watermarking: Detect

---

1: **Input:** Suspect text $y_{1:n}$, watermark key k, pseudo-random function $F$, target false positive rate $\alpha$
2: **Output:** Binary decision (1 if text is watermarked, 0 otherwise)
3: Calculate the cumulative score

$$\text{TestScore}_{\text{PF}}(y_{1:n}) = \sum_{t=m+1}^{n} -\log(r_t(y_t)) \tag{7}$$

   where $r_t(y) = F_{y_{t-m:t-1}, \mathsf{k}}(y)$
4: **if** $\text{TestScore} > \text{CDF}^{-1}_{\text{Gamma}(n-m,1)}(1 - \alpha)$ **then return** 1, i.e., "The suspect text is watermarked."
5: **else return** 0, i.e., "The suspect text is not watermarked."

---

**Theorem 4.3.** *Assume the pseudo-randomness is perfect[2], i.e., $F_{w_{1:m}, \mathsf{k}}(y) \sim \text{Uniform}([0, 1])$ i.i.d. $\forall [w_{1:m}, y] \in \mathcal{V}^{m+1}$.*

*The following are true about PF watermark scheme.*

1. *If $y_{1:n}$ is* statistically independent *to the secret key* k*,*

$$\mathbb{E}\left[\text{TestScore}_{\text{PF}}(y_{1:n}) | y_{1:n}\right] = n - m.$$

2. *If in addition, all $m$-grams in $y_{1:n}$ are unique, then conditioning on $y_{1:n}$,*

$$\text{TestScore}_{\text{PF}}(y_{1:n}) \sim \text{Gamma}(n - m, 1).$$

   *The choice $\tau = \text{CDF}^{-1}_{\text{Gamma}(n-m,1)}(1 - \alpha)$ ensures the false positive rate in Algorithm 3 is equal to $\alpha$.*
3. *Assume $y_{1:n}$ is drawn from Algorithm 2, then*

$$\mathbb{E}\left[\text{TestScore}_{\text{PF}}(y_{1:n})\right] = \sum_{t=m+1}^{n} \mathbb{E}\left[\sum_{y \in \mathcal{V}} \int_{0}^{e^{u_t(y)-u_t^*}} \left( -\log r \cdot \prod_{y' \in \mathcal{V}, y' \neq y} \left(1 - r \cdot e^{u_t(y')-u_t(y)}\right) \right) dr \right]. \tag{8}$$

The above expression in (8) may appear messy, but it is the exact calculation and captures the entropy of the distribution PF-induces for a given $u_t$. To see this, let us instantiate the bound for two special cases that admit more explicit forms.

**Example 4.4.** When $\text{Softmax}(u_t)$ is $1/k$ for an arbitrary subset of $k$ tokens and 0 for others,

$$\mathbb{E}[-\log(r_t(y_t))] := H_k = 1 + 1/2 + ... + 1/k \approx \log k.$$

Specifically, when $k = |\mathcal{V}|$ this is the uniform distribution, $(8) \asymp n \log |\mathcal{V}|$ while when $k = 1$, the sequence is completely deterministic (e.g., when the LLM is asked to recite the "Declaration of Independence"), then we get $(8) = n - m$ as expected.

In the above example, (8) is identical to the expected TestScore of the Gumbel watermark in (3). This is because the distributions they sample from are also the same. To illustrate their difference, let us revisit the simple two-token case from Example 3.2 again for which we can work out the expectation of the test score explicitly.

---

[2]This is a simplifying assumption. We only need $(n - m)|\mathcal{V}|$-way independence.

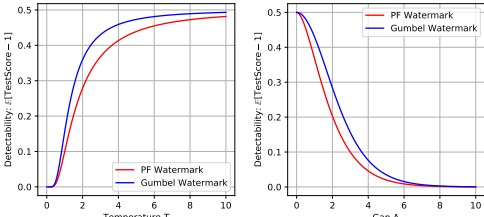 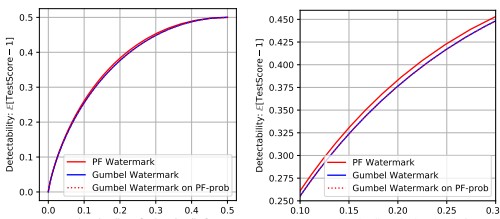

(a) Comparing the *detectability* of PF watermark vs Gumbel watermark using Example 4.5. On the left, we fix the Gap $\Delta = 3.0$ and vary $T$. On the right, we fix $T = 1.0$ and vary $\Delta$. Gumbel watermark offers higher detectability when $T$ is the same. **Very importantly** this does *not* mean Gumbel outperforms PF because PF is more greedy and has less entropy.

(b) Comparing the *detectability-greediness* tradeoff of PF watermark vs Gumbel watermark in the two token case. The Gap $\Delta = 3.0$, both curves are traced out by varying the temperature $T$ – with a "zoomed-in" view on the RHS. It is clear from the figure that PF outperforms Gumbel on the tradeoff curve.

Figure 2: Comparing the *detectability* of PF watermark vs Gumbel watermark using Example 4.5.

**Example 4.5.** Let the $|\mathcal{V}| = 2$ and the corresponding logits be $[\Delta, 0]$. The expected TestScore of the Gumbel and the PF watermark (for each watermarked token) are: $\frac{H_{1+e^{-\Delta/T}}}{1+e^{-\Delta/T}} + \frac{H_{1+e^{\Delta/T}}}{1+e^{\Delta/T}}$ and $1 + \frac{1}{2}e^{-\Delta/T}(1 + \Delta/T)$ respectively, where $H_x$ is the $x^{th}$ Harmonic number.

It is a bit hard to compare them by reading the mathematical expressions, so let us compare them numerically (see Figure 2a). The vertical axis in the figures measures *Detectability*, which we define to be the expected difference between the TestScore of a watermarked and unwatermarked token. Since under the null the $\mathbb{E}[-\log(r_t(y_t))] = E[-\log(1 - r_t(y_t))] = 1$, we can simply subtract 1 from the expressions in Example 4.5.

Figure 2a indicates the PF watermark does not beat the Gumbel watermark in terms of *detectability* when $T$ is fixed. This should not be surprising since for the same temperature, PF watermark is better at optimizing (recall Example 3.2 and Figure 1), thus naturally the resulting distribution has less entropy to be exploited by the watermarking scheme.

A more fair comparison, would be to increase the temperature for PF watermark appropriately so we compare *detectability* when the *suboptimality* is aligned. This is shown in Figure 2b. In fact we have added a second baseline that apply Gumbel watermark to the *induced sampling distribution from PF-decoding* (shown as the dotted line). The distribution induced by PF does not have a simple form, but in our special case, it was worked out in Example 3.2.

As we can see in Figure 2b, the PF watermark is never worse and even has a slight advantage in the middle. To say it differently, to achieve the same suboptimality, the PF watermark can afford to use a larger temperature, which not only improves the stability parameter but also compensates it sufficiently on the detectability front to outperform the Gumbel watermark. In practice, we expect PF watermark to be as effective as the Gumbel watermark, and could even be a bit better (if the temperature parameter is chosen appropriately).

In conclusion, we showed that the watermarked version of PF-decoder is computationally indistinguishable from the original version of PF-decoder. Meanwhile, the test score of the PF watermark is qualitatively similar to that of the Gumbel-watermark (and identical in some cases). It is likely to produce similar detectability to the Gumbel watermark, while enjoying the performance boost that comes from replacing softmax sampling with PF.

## 5 EXPERIMENTS

**Datasets and models.** We utilize two long-form text datasets in our experiments: the Colossal Clean Crawled Corpus (C4) dataset (Raffel et al., 2020) for open-ended text completion generation, and the Alpaca dataset (Taori et al., 2023) for question-answering tasks. Our primary language model is the state-of-the-art open-source model Llama-2 with 7 billion parameters. Specifically, we use the Llama-2-7B-chat model for question-answering tasks on the Alpaca dataset. For text

Table 2: Text generation results for different methods. The true positive rate (TPR) is calculated under 0.01 false positive rate (FPR). PPL1 refers to perplexity measured by Llama2-7B models. PPL2 is perplexity calculated by the Llama2-13B model. For general text generation, PF decoding produces significantly lower perplexity compared to sampling. For watermarking methods, PF watermark also produces lower perplexity compared to KGW watermark and Gumbel watermark.

| Method | AUC↑ | TPR↑ | PPL1↓ | PPL2↓ | Seq-rep-5↓ | MAUVE↑ | Method | AUC↑ | TPR↑ | PPL1↓ | PPL2↓ | Seq-rep-5↓ | MAUVE↑ |
|---|---|---|---|---|---|---|---|---|---|---|---|---|---|
| **C4, T=1.0, Llama2-7B** | | | | | | | **C4, T=0.8, Llama2-7B** | | | | | | |
| Greedy | - | - | $1.14_{0.01}$ | $1.24_{0.03}$ | 0.56 | 0.05 | Greedy | - | - | $1.28_{0.02}$ | $1.75_{0.03}$ | 0.12 | 0.93 |
| Sampling | - | - | $12.47_{0.32}$ | $15.31_{0.41}$ | 0.02 | 0.98 | Sampling | - | - | $4.23_{0.06}$ | $4.91_{0.08}$ | 0.06 | 1.00 |
| PF | - | - | $8.94_{0.20}$ | $10.75_{0.25}$ | 0.03 | 0.90 | PF | - | - | $3.54_{0.06}$ | $4.11_{0.08}$ | 0.10 | 0.92 |
| KGW WM | 0.989 | 0.991 | $16.62_{0.38}$ | $20.62_{0.49}$ | 0.01 | 1.00 | KGW WM | 0.995 | 0.991 | $5.78_{0.08}$ | $6.77_{0.11}$ | 0.03 | 0.99 |
| Gumbel WM | 0.997 | 0.988 | $11.41_{0.27}$ | $14.12_{0.36}$ | 0.04 | 0.93 | Gumbel WM | 0.995 | 0.982 | $4.03_{0.07}$ | $4.71_{0.09}$ | 0.10 | 1.00 |
| PF WM | 0.995 | 0.984 | $8.33_{0.20}$ | $10.28_{0.29}$ | 0.05 | 0.99 | PF WM | 0.993 | 0.980 | $3.38_{0.07}$ | $3.99_{0.10}$ | 0.13 | 1.00 |
| **Alpaca, T=1.0, Llama2-7B-Chat** | | | | | | | **Alpaca, T=1.0, TinyLlama-1.1B-Chat** | | | | | | |
| Greedy | - | - | $1.28_{0.02}$ | $1.75_{0.03}$ | 0.12 | 0.93 | Greedy | - | - | $1.41_{0.01}$ | $1.66_{0.02}$ | 0.30 | 0.99 |
| Sampling | - | - | $1.74_{0.02}$ | $2.41_{0.04}$ | 0.09 | 0.86 | Sampling | - | - | $2.73_{0.04}$ | $3.71_{0.06}$ | 0.11 | 1.00 |
| PF | - | - | $1.65_{0.02}$ | $2.30_{0.04}$ | 0.09 | 0.98 | PF | - | - | $2.53_{0.03}$ | $3.44_{0.06}$ | 0.12 | 0.98 |
| KGW WM | 0.961 | 0.596 | $2.20_{0.04}$ | $3.00_{0.06}$ | 0.08 | 0.93 | KGW WM | 0.998 | 0.991 | $3.81_{0.06}$ | $5.28_{0.09}$ | 0.07 | 0.99 |
| Gumbel WM | 0.986 | 0.858 | $1.70_{0.02}$ | $2.35_{0.04}$ | 0.10 | 0.93 | Gumbel WM | 1.000 | 0.995 | $2.67_{0.04}$ | $3.58_{0.06}$ | 0.12 | 1.00 |
| PF WM | 0.979 | 0.810 | $1.69_{0.03}$ | $2.37_{0.04}$ | 0.10 | 1.00 | PF WM | 0.999 | 0.986 | $2.36_{0.04}$ | $3.15_{0.07}$ | 0.14 | 0.94 |

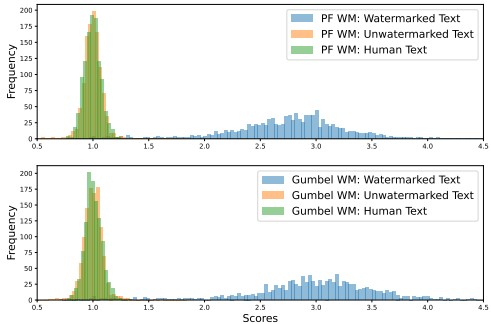

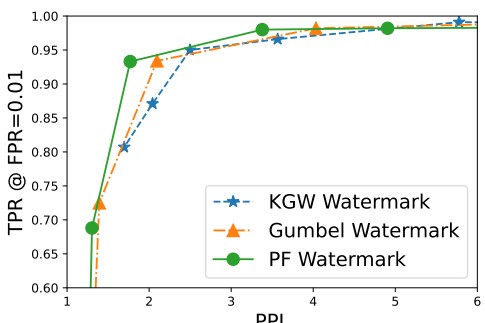

(a) TestScore distribution. We calculate the average TestScore of the PF watermark and Gumbel watermark using Llama2-7B (T=1.0) on the C4 dataset. The length of the suspect texts is fixed at 200 tokens. A clear gap emerges between positive samples (watermarked) and negative samples (unwatermarked and human-written), indicating the watermark detectability.

(b) Trade-off between detection accuracy (TPR at FPR=0.01) and text quality (PPL) across three watermark configurations on the C4 dataset, with temperature settings ranging from 0.2 to 1.0. Each data point represents the outcome for 500 watermarked texts. The PF watermark achieves the optimal balance of the highest detection accuracy and lowest perplexity.

Figure 3: Comparison of PF and Gumbel watermarks on real data.

completion tasks on the C4 dataset, we employ the base model Llama-2-7B. Furthermore, to evaluate the universal applicability of smaller models, we also assess the performance of the TinyLlama-1.1B model[3] (Zhang et al., 2024).

**Evaluation metrics.** We calculate perplexity scores from different models, using Llama2-7B to compute PPL1 and Llama2-13B to compute PPL2. We also compute MAUVE scores to measure the distributional similarity between model generations and human text as another metric for text quality (Pillutla et al., 2021). To evaluate repetitiveness, we compute seq-rep-5 across generations, which is the average repetition rate of duplicate 5-grams in a sequence (Welleck et al., 2020). For the watermark evaluation, maintaining a low false positive rate is crucial to avoid misclassifying unwatermarked text as watermarked. Therefore, we set the false positive rates at 1% and 10% for all watermark detection algorithms, adjusting the detection threshold accordingly. We report true positive rate (TPR) and F1 scores to measure the watermark detectability. We compared the well-known Gumbel Watermark (Gumbel WM) and Green-Red Watermark (KGW WM) as our main baselines. Experiments were conducted using Nvidia A6000 GPUs. For the details of the experiment setting, please refer to the Appendix C.

---

[3] https://huggingface.co/TinyLlama/TinyLlama-1.1B-Chat-v1.0

**Text generation performance.** Table 2 shows the text perplexity of generated samples from different LLMs evaluated on two datasets. Using the same temperature, we find that PF decoding produces significantly lower perplexity compared to sampling. Although greedy decoding has the lowest perplexity, it suffers from heavy repetition, as indicated by its high seq-rep-5 score and low MAUVE score. We observe that for question-answering tasks, the perplexity is lower, likely due to the fixed form of answers and lower entropy of the text generation. Table 8 shows an example prompt and responses generated by different decoding methods.

**Watermarking results.** We compare the results of PF watermark with those of the Gumbel Watermark (Gumbel WM) and the Green-Red watermark (KGW WM). In Figure 3a, we present the distribution of detection scores for the PF watermark. The PF watermark demonstrates clear detectability between positive and negative samples. The results of the watermark generation are shown in Table 2 and Figure 3b. The PF watermark achieves the best balance of the highest detection accuracy and lowest perplexity, compared to the KGW WM and the Gumbel WM. Notably, the perplexity of the PF watermark is close to that of the PF sampling, indicating that the watermarking process does not significantly impact the quality of the generated text. All watermarking methods achieved near-perfect detection accuracy on the C4 dataset. Besides, the detection results for the small TinyLlama model are also good, demonstrating the universal applicability of the PF watermark.

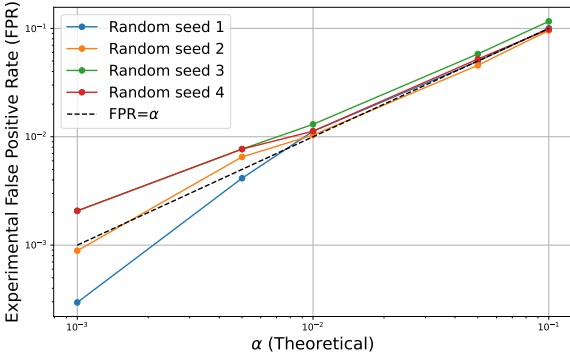

Figure 4: Comparison of empirical and theoretical false positive rates with different watermark keys. We can see that the second statement of Theorem 4.3 correctly controls the Type I error in practice.

**Controlling the false positive rate.** The key strength of PF watermark is its ability to precisely control the false positive rate (FPR) during detection. We validate this by conducting experiments using negative examples from diverse datasets (C4, Alpaca, unwatermarked) and different random keys. As Figure 4 shows, the empirical false positive rates align tightly with the theoretical $\alpha$ values across different settings. This demonstrates PF watermark's effectiveness in precisely controlling the FPR as intended.

**Additional watermarking results.** For a text watermarking design to be effective, it should be able to withstand paraphrasing attacks that an adversary may attempt to modify the watermarked text. Furthermore, the watermark should be detectable even with shorter text lengths. In Appendices C.1.1 and C.2.2, we present additional empirical results for the PF watermark, demonstrating its robustness to paraphrasing and editing attacks. The results also show that the PF watermark can still be detected even when the length of the text is reduced to only 30 tokens.

## 6 CONCLUSION

We introduce Permute-and-Flip (PF) decoding, a new decoding method for large language models that enjoys the same – perturbation-stability guarantees as softmax sampling while achieving substantially lower perplexity. We design a tailored watermarking scheme (PF watermark) for PF decoding that enables precise control over false positive rates while retaining high true positive rates. Our experiments demonstrate that the PF watermark achieves the best balance of the highest detection accuracy and lowest perplexity. All these intriguing properties make PF decoding a promising new approach for practical applications of large language models.

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

## A    MORE ON RELATED WORK

### A.1    LANGUAGE MODEL DECODING.

The decoding strategy used in text generation greatly impacts the resulting text's quality and diversity. Traditional deterministic algorithms, like greedy decoding and beam search, often lead to repetitive text (Wiher et al., 2022). To address this, diverse beam search (DBS) (Vijayakumar et al., 2016) has been developed to promote diversity in text generation. Stochastic decoding strategies, such as Top-$k$ and Top-$p$ (Nucleus) (Holtzman et al., 2020) sampling, balance randomness and determinism, selecting from the most likely tokens to enhance variety while maintaining coherence. The Bayes Minimum Risk (MBR) method minimizes expected risk and incorporates a utility function to navigate trade-offs between text attributes. Advanced techniques have been developed to improve decoding for large language models, including the imposition of constraints (Anderson et al., 2017; Qin et al., 2020; Hokamp & Liu, 2017; Lu et al., 2022), enhancing text quality (Li et al., 2023), and speeding up the decoding process (Chen et al., 2023a).

Our contributions are complementary to these existing methods in that we are the first to introduce a rigorous stability definition and study the tradeoff between utility (e.g. perplexity) and stability. Permute-and-flip sampling can be used as a drop-in replacement for softmax sampling whenever it is used, e.g., in standard full sampling or nucleus (Top-p) sampling. We also provide watermarking capabilities for PF-decoder. We believe that the PF decoder has the potential to become a promising new approach for language model decoding.

### A.2    DETECT AI-GENERATED TEXT

Another major motivation of the work is to come up with a reliable method for detecting AI-generated text, so as to prevent LLM misuse. We briefly review two categories of existing work on this problem.

**Post-hoc detection.**    Post-hoc detection of LLM-generated text encompasses two main approaches: zero-shot detection and training-based detection. Zero-shot detection is characterized by its capacity to identify AI-generated text without needing specific training data, leveraging the inherent stylistic differences between human and machine writing. Techniques within this category, such as DetectGPT (Mitchell et al., 2023), PHD (Tulchinskii et al., 2023), DNA-GPT (Yang et al., 2024), and Fast-DetectGPT (Bao et al., 2024), utilize metrics like log-probability scores, n-gram frequencies, lower intrinsic dimensionality, and conditional probability to differentiate AI-generated content. In contrast, training-based detection involves fine-tuning pre-trained language models on datasets that consist of both human and LLM-generated texts to build a classifier. This method is exemplified by various systems, including commercial detection platforms (OpenAI, 2023a; GPTZero, 2023; ZeroGPT, 2023), and research projects (Chen et al., 2023b; Yu et al., 2023; Liu et al., 2023; Hu et al., 2023), which leverage the capabilities of large language models to effectively classify text origins. However, despite post-hoc detection's effectiveness in many cases, recent studies show detection methods' robustness is limited across different scenarios. They have proven fragile to adversarial attacks and biased against non-native English writers (Wolff, 2020; Sadasivan et al., 2023; Liang et al., 2023; Shi et al., 2024). Limitations in accuracy even led OpenAI to close their detector in July 2023 (OpenAI, 2023a).

**LLM watermarking.**    The watermarking approach provides a direct solution for AI text detection by intentionally embedding detectable signals or "watermarks" within the text. Unlike post-hoc detection, watermarking aims to determine if the text originates from a specific language model and it is robust to distribution shifts. Evolving from earlier techniques such as synonym substitution (Topkara et al., 2006) and syntactic restructuring (Atallah et al., 2001), modern watermarking strategies involve integrating watermarks into the decoding process of language models (Zhao et al., 2023; Kirchenbauer et al., 2023). Aaronson (2023) works with OpenAI to first develop a Gumbel watermark that uses a "traceable" pseudo-random softmax sampling when generating the next word. Kirchenbauer et al. (2023) split the vocabulary into red-green lists based on hash values of previous n-grams and then increase the logits of green tokens to embed the watermark. Zhao et al. (2024a) provides strong theoretical guarantees for the green-red watermarks and advocates the use of a consistent red-green list to enhance robustness to evasion attacks. Christ et al. (2024); Hu et al. (2024); Kuditipudi et al. (2024); Wu et al. (2024) study watermarks that preserve the original token probability distributions.

Meanwhile, multi-bit watermarks (Yoo et al., 2023; Fernandez et al., 2023) have been proposed to embed more complex information in the generation tasks.

PF-watermark is a newcomer to the family of LLM watermarks. It is closest to the Gumbel watermark (Aaronson, 2023) and enjoys all desirable properties of the Gumbel watermark. In Section 4 we have thoroughly compared the two watermarks with theory and numerical simulation, demonstrating that PF-watermarks offer a slightly improved detectability-greedness tradeoff. Comparisons under real-data experiments were also presented in Section 5.

Our results also have interesting implications for the green-red watermark (Kirchenbauer et al., 2023). For example, we can consider a PF-sampling version of green-red watermark which may perform better than the current green-red watermark in terms of its detectability-distortion tradeoff. Specifically, the stability guarantee of the PF-decoder (Theorem 3.1 Statement 1) implies that PF Green-Red Watermark enjoys the same quality guarantee as is analyzed in (Zhao et al., 2024a, Theorem 3.1) for $\alpha = \infty$, but the more concentrated distribution might make the watermark more prominent / and more detectable for PF-sampling based Green-Red watermark. A full exploration of this idea will be presented in future work or a longer version of the current paper.

### A.3 Differential Privacy

While the current work is not about privacy, our technical approach heavily relies on existing methods developed in the differential privacy literature. Specifically, the permute-and-flip sampling was developed by McKenna & Sheldon (2020) as an alternative to the classical exponential mechanism (McSherry & Talwar, 2007) for the problem of differentially private selection. Ding et al. (2021) shows that the PF sampling is equivalent to returning the argmax of a noisy version of the utility function perturbed by independent exponential noise. Moreover, stability can be viewed as an algorithm-centric, input-specific version of pure-differential privacy (Dwork et al., 2006).

While some of the results we present are directly implied by existing work (e.g., Theorem 3.1), we believe it is a worthwhile (and hopefully beneficial) effort to introduce these results and their implications to the broader machine learning community.

To our knowledge, we are the first to draw the connection between various versions of Report-Noisy-Max (RNM) mechanisms in differential privacy to the LLM watermarking problem. Besides Gumbel noise (Gumbel-watermark) and exponential noise (PF-watermark), there are other versions of RNM that add, e.g., Gaussian noise (Zhu & Wang, 2022, Corollary 9). We hope our work could inspire more interplay between these two research areas.

## B  More Discussion of Stability

In this section, we delve deeper into the concept of stability as defined in Definition 2.1.

### B.1 Stability and Its Implication for Diversity

The definition of stability implies that the LLM decoder is resistant to small perturbations in the logits. Furthermore, *stability* implies an intuitive notion of *diversity*, which says that for tokens with similar logits, then their chances of getting chosen should be similar. More rigorously:

*Remark* B.1 (Stability implies diversity). If $|u_t(y) - u_t(y')| \leq \delta$, then we can construct a $\tilde{u}_t$ such that $\tilde{u}_t(y) = \tilde{u}_t(y')$ while satisfying $\|u_t - \tilde{u}_t\|_\infty \leq \frac{\delta}{2}$. Apply triangle inequality and Definition 2.1, we get

$$\left| \log \frac{p_{\mathcal{A}_{u_t}}(y)}{p_{\mathcal{A}_{u_t}}(y')} \right| = \left| \log \frac{p_{\mathcal{A}_{u_t}}(y)}{p_{\mathcal{A}_{\tilde{u}_t}}(y)} + \log \frac{p_{\mathcal{A}_{\tilde{u}_t}}(y')}{p_{\mathcal{A}_{u_t}}(y')} \right| \leq L\delta.$$

### B.2 Comparison of Decoding Methods in Terms of Stability

Theorem 3.1 establishes the Pareto-optimality of PF sampling with respect to the stability-perplexity tradeoff. This implies that no other decoding algorithm can surpass PF sampling in both stability and perplexity simultaneously.

Table 3: Comparison of stability and suboptimality across different sampling methods.

| Property | Softmax | PF | Beam/Greedy/Top-$k$/Top-$p$ |
|---|---|---|---|
| Stability (Def 2.1) | $L = 2/T$ | $L = 2/T$ | $+\infty$ (Not stable for any parameter $L$) |
| Expected Suboptimality (Worst-case bound) | $T \log|V|$ | $T \log|V|$ | 0 for "greedy"; $T \log k$ for "Top-$k$" |
| Expected Suboptimality (Per instance) | Exponentially higher log-likelihood | Between 1 - 2 smaller than Softmax | 0 for "greedy"; Same as Softmax on "Top-$k$" |

Table 3 summarizes the stability guarantees of various decoding methods. Notably, only softmax sampling and PF sampling exhibit provable stability according to Definition 2.1. This is in contrast to other popular methods like beam search, greedy search, and Top-$k$/Top-$p$ sampling, which lack such guarantees.

In our definition, stability implies that small perturbations to logits should not significantly alter the output text distribution. This definition also constrains the degree of greediness that a decoder can exhibit, as small changes to logits may change the sorted order of the next token. Thus, it is clear why greedy decoding and Top-$k/p$ sampling cannot be stable.

When we say an algorithm $\mathcal{A}$ cannot be stable, we mean that for any $L < +\infty$, there exist two decoding problems $u_t$ and $\tilde{u}_t$ such that $|u_t - \tilde{u}_t|_\infty \le \delta$ for some $\delta > 0$, but the log-probability ratio between $A(u_t)$ and $A(\tilde{u}_t)$ exceeds $\delta L$.

To establish non-stability, we only need a counterexample. Below, we construct explicit counterexamples for each decoding method:

- Greedy: We have a vocabulary of two tokens. Let $u_t = [0, -1]$ and $\tilde{u}_t = [1, 0]$. This satisfies that $|u_t - \tilde{u}_t|_\infty \le \delta$ with $\delta = 1$. But Greedy($u_t$) outputs the second token with probability 1, Greedy($\tilde{u}_t$) outputs the second token with probability 0. Therefore, the importance ratio is unbounded, i.e., there isn't any finite $L$ that bounds the probability ratio.
- Top-$k$: Let's consider $k = 2$ as an example, and let vocabulary size be 3. Let $u_t = [0, -1, -2]$ and $\tilde{u}_t = [0, -2, -1]$. Again, $|u_t - \tilde{u}_t|_\infty \le \delta$ with $\delta = 1$. Now, Top-2 sampling will never output the third token on $u_t$, while it will output the third token with probability $> 0$ on $\tilde{u}_t$. The importance ratio is unbounded.
- Top-$p$: Same example as in Top-$k$ above. The softmax of $u_t$ and $\tilde{u}_t$ are approximately $[0.844, 0.114, 0.042]$ and $[0.844, 0.042, 0.114]$ respectively. If we set $p = 0.95$, then Top-$p$ with $u_t$ will exclude the third token, while that with $\tilde{u}_t$ will retain it — leading to an importance ratio of $\infty$.
- Beam Search: Since beam search with a constant score function reduces to greedy decoding, the same counter-example for greedy decoding demonstrates its non-stability.

Therefore, these counter-examples highlight that greedy, Top-$k$, Top-$p$, and beam search decoding are not stable according to our definition.

### B.3 The Connection Between Stability and Safety

Stability can provide robustness against certain types of adversarial attacks, such as jailbreaking, within a gray-box threat model.

Consider a scenario where an attacker can alter the logits and receive responses from an API, as is common with OpenAI's logit bias feature[4]. Recent research has shown that it is possible to jailbreak an LLM to encourage specific outputs like "Sure" or "Definitely" or other tokens (Zhang et al., 2023; Zhao et al., 2024b). By employing the PF decoding method, the API provider can make jailbreaking significantly harder while also making the output watermarkable, thereby enhancing the model's safety and security.

---

[4]https://help.openai.com/en/articles/5247780-using-logit-bias-to-alter-token-probability-with-the-openai-api

Stability serves as a fundamental guarantee against these perturbations. That said, all attacks will eventually manifest as perturbations to logits, and having a "Lipschitz" transformation from logits to token probabilities is a nice guarantee. Without this, even tiny changes to the logits can have a big influence on the final token probability distribution.

For instance, consider the difference between a provably stable decoder (such as softmax) and a non-stable one in a gray-box jailbreaking scenario. Suppose the language model has a vocabulary of just three words: "Sure", "OK", and "Sorry". Initially, the logits $u_t = [-2, -1, 0]$ indicate that the LLM is inclined to output "Sorry". If the adversary aims to increase the likelihood of outputting "Sure", they could perturb the logits to $\tilde{u}_t = [0, -1, 0]$, which satisfies $|u_t - \tilde{u}_t|_\infty \leq \delta$ with $\delta = 2$.

When applying Top-$k$ sampling with $k = 2$ (as an example of several sampling methods that lack stability, such as Greedy, Top-$k$, Top-$p$, and Beam Search), the model would never select "Sorry", the third token, whereas with the original logits, "Sorry", would still have a non-zero probability of being selected. This demonstrates that Top-$k$ sampling lacks *stability*. In contrast, PF sampling and softmax sampling maintain *stability* under such perturbations.

## C  ADDITIONAL EXPERIMENT DETAILS

We provide additional details on the experiments here. We use the C4 (Raffel et al., 2020) and Alpaca (Taori et al., 2023) datasets. Specifically, we select samples from C4 with text longer than 500 tokens, using the first 200 tokens as the prompt input to the language model and the next 300 tokens as the human-generated reference. This gives us 600 examples. For Alpaca, we select samples with prompts/instructions longer than 25 tokens and answers also longer than 25 tokens, giving 550 examples. Since Llama2-Chat is a fine-tuned version of Llama-2 optimized for dialogue, we use the Chat version (Llama-2-7B-Chat[5]) for the question-answering task and the base model (Llama-2-7B[6]) for the text completion task.

Given that PF decoding can integrate with Top-$p$ sampling, which initially selects the top $p$ tokens before normalization, we conduct the performance tests using a $p = 1.0$ for full sampling. The max generation length is set to 256 tokens for all experiments.

For perplexity calculation, we observe high variance with different methods, often influenced by outliers. To address this, we remove the top and bottom 3% of perplexity scores as outliers and then calculate the average perplexity and standard error. For MAUVE scores, we use the human-written references from C4 and Alpaca as the human distribution.

For watermarking experiments, we generate 500 watermarked and 500 unwatermarked sentences per method. We label them as "watermarked" and "unwatermarked" respectively, with corresponding human-written text as "human" for each prompt. Following Kirchenbauer et al. (2023), we use a watermark strength of $\delta = 2.0$ and green list ratio of $\gamma = 0.5$ for the KGW watermark. For fair comparison, we use the same long prefix as the pseudo-random function, hashing the previous m tokens to get the random vector for Gumbel/PF watermarks, or to split the green/red token lists. For the watermark robustness test (Table 4) we use a 4-token prefix, and an 8-token prefix elsewhere. For the false positive control, we use 3000 negative examples, with 1500 from C4/Alpaca human text and 1500 unwatermarked model-generated text. In our robustness testing, we evaluate two configurations of the DIPPER (Krishna et al., 2023) model: DIPPER-1 with lexical diversity $L = 40$, order diversity $O = 40$, and DIPPER-2 with $L = 40, O = 100$.

### C.1  PF WATERMARK ROBUSTNESS RESULTS.

To evaluate the robustness of the watermark detection, we test the PF watermark under paraphrasing and text editing attacks. Note that robustness here refers to the watermark's resilience against removal attacks, which is different from the logit stability in Definition 2.1.

We employ various paraphrase attack techniques intended to remove the watermark text. The experiments are conducted with a 4-token prefix for the pseudorandom function. We also added the

---

[5]https://huggingface.co/meta-llama/Llama-2-7b-chat-hf
[6]https://huggingface.co/meta-llama/Llama-2-7b-hf

Table 4: Detection results for three watermarking methods using Llama2-7B on the C4 dataset under different attacks.

| Setting | Method | AUC | 1% FPR | | 10% FPR | |
|---|---|---|---|---|---|---|
| | | | TPR | F1 | TPR | F1 |
| No attack | KGW (m=4) | 0.998 | 0.996 | 0.989 | 1.000 | 0.906 |
| | KGW (m=1) | 0.999 | 1.000 | 0.995 | 1.000 | 0.906 |
| | Gumbel | 0.992 | 0.979 | 0.979 | 0.986 | 0.913 |
| | PF | 0.996 | 0.977 | 0.980 | 0.993 | 0.898 |
| DIPPER-1 | KGW (m=4) | 0.661 | 0.057 | 0.105 | 0.317 | 0.416 |
| | KGW (m=1) | 0.876 | 0.389 | 0.554 | 0.717 | 0.720 |
| | Gumbel | 0.838 | 0.367 | 0.529 | 0.642 | 0.697 |
| | PF | 0.824 | 0.374 | 0.537 | 0.622 | 0.684 |
| DIPPER-2 | KGW (m=4) | 0.638 | 0.051 | 0.096 | 0.278 | 0.375 |
| | KGW (m=1) | 0.885 | 0.342 | 0.501 | 0.662 | 0.714 |
| | Gumbel | 0.764 | 0.239 | 0.380 | 0.523 | 0.608 |
| | PF | 0.795 | 0.250 | 0.394 | 0.544 | 0.625 |
| Random Delete (0.3) | KGW (m=4) | 0.936 | 0.484 | 0.644 | 0.881 | 0.844 |
| | KGW (m=1) | 0.956 | 0.752 | 0.836 | 0.923 | 0.839 |
| | Gumbel | 0.981 | 0.941 | 0.960 | 0.959 | 0.898 |
| | PF | 0.985 | 0.936 | 0.956 | 0.966 | 0.888 |

original soft watermarking method with a 1-token prefix for comparison ($m = 1$ for KGW watermark (Kirchenbauer et al., 2023)).

In our experiments, we utilize two versions of the DIPPER paraphrasing model (Krishna et al., 2023), denoted as DIPPER-1 and DIPPER-2. DIPPER-2 generates more diverse paraphrases than DIPPER-1. Moreover, we test a random word deletion attack, which is a common technique used to manipulate text. These experiments represent realistic scenarios where an adversary may attempt to remove watermarks through paraphrasing or editing. The results, shown in Table 4, illustrate the robustness of the PF watermark to these paraphrasing and editing attacks. The PF watermark achieves comparable detection performance to the Gumbel watermark and KGW watermark methods when using the same long prefix as the pseudorandom function.

Table 5: PF watermark detection results with different lengths.

| Length | AUC | 1% FPR | | 10% FPR | |
|---|---|---|---|---|---|
| | | TPR | F1 | TPR | F1 |
| 200 | 0.994 | 0.977 | 0.978 | 0.985 | 0.915 |
| 150 | 0.993 | 0.975 | 0.980 | 0.985 | 0.913 |
| 100 | 0.992 | 0.970 | 0.972 | 0.983 | 0.911 |
| 50 | 0.987 | 0.950 | 0.966 | 0.970 | 0.902 |
| 30 | 0.980 | 0.923 | 0.950 | 0.953 | 0.888 |

## C.2 IMPACT OF TEXT LENGTH ON WATERMARK DETECTION.

Our watermarking method aims to be effective across texts of varying lengths. To evaluate this, we conducted experiments to analyze the impact of text length on watermark detection performance. Texts are truncated to 30, 50, 100, 150, and 200 tokens. The results, shown in Table 5, validate the robustness of our approach to different text lengths. Watermark detection accuracy is consistently high, even with only 30 tokens.

Table 6: KGW Watermark with $\gamma = 0.1$, $\delta = 2.0$ and single token for hash

| Temperature | TPR@0.01FPR | PPL1 |
|---|---|---|
| 1.0 | 0.690 | 15.07 |
| 0.8 | 0.686 | 4.434 |
| 0.6 | 0.681 | 2.047 |
| 0.4 | 0.643 | 1.768 |
| 0.2 | 0.628 | 1.559 |

Table 7: KGW Watermark with $\gamma = 0.25$, $\delta = 2.0$ and single token for hash.

| Temperature | TPR@0.01FPR | PPL1 |
|---|---|---|
| 1.0 | 0.973 | 16.31 |
| 0.8 | 0.972 | 4.887 |
| 0.6 | 0.934 | 2.328 |
| 0.4 | 0.893 | 1.940 |
| 0.2 | 0.859 | 1.791 |

### C.3 EXPERIMENTS ON VARIANTS OF KGW WATERMARK

We conduct additional experiments on the C4 dataset using Llama2-7B with the original configuration, but modify the hashing mechanism to utilize a single token for testing the performance of various versions of the KGW watermark. We select $\gamma$ values of 0.25 and 0.1. The results can be found in Tables 6 and 7. Upon comparing these results with the data from Figure 3b, the PF watermark remains at the Pareto front, providing the optimal trade-off between the highest detection accuracy and the lowest perplexity.

## D   PROOFS OF TECHNICAL RESULTS

### D.1   PERMUTE AND FLIP SAMPLING

First, let us calculate the probability of Permute-and-Flip sampling from Line 3-9 of Algorithm 1. We will use the equivalent ReportNoisy(Arg)Max form from Fact 4.2.

$$w_t = \arg\max_{w \in \mathcal{V}} \left( u_{w,t} - \log r_{w,t} \right)$$

First, observe that the event that "$w$ is selected" is the same as the event that for $u_w - \log r_w > u_{w'} - \log r_{w'}$ for all $w' \neq w$.

Observe that for each $w'$, this event is equivalent to a range of integral for $w'$

$$u_w - \log r_w > u_{w'} - \log r_{w'} \Leftrightarrow \log r_{w'} > -u_w + u_{w'} + \log r_w \Leftrightarrow r_{w'} > r_w e^{u_{w'} - u_w} \quad (9)$$

We have

$$\Pr[w \text{ is selected}] = \mathbb{E}\left[\mathbf{1}\left(w \text{ is selected}\right)\right]$$

$$= \int_0^1 \prod_{w' \neq w} \left(\int_0^1 \mathbf{1}\left(u_w - \log r_w > u_{w'} - \log r_{w'}\right) \mathrm{d}w'\right) \mathrm{d}r_w$$

$$= \int_0^1 \prod_{w' \neq w} \left(\int_{r_w \exp\left(u_{w'} - u_w\right)}^1 \mathrm{d}r_{w'}\right) \mathrm{d}r_w$$

$$= \int_0^1 \prod_{w' \neq w} \left(1 - r_w \cdot e^{u_{w'} - u_w}\right)_+ \mathrm{d}r_w$$

$$= \int_0^{e^{u_w - u_{w^*}}} \prod_{w' \neq w} \left(1 - r_w \cdot e^{u_{w'} - u_w}\right) \mathrm{d}r_w \tag{10}$$

$$= \int_0^{\frac{p(w)}{p(w^*)}} \prod_{w' \neq w} \left(1 - r_w \cdot \frac{p(w')}{p(w)}\right) \mathrm{d}r_w$$

where $(x)_+ := \max(0, x)$, and $p(\cdot) := \text{Softmax}(u)$. In the above, $w^* = \arg\max_w u_w$, and observe that

- If $w = w^*$, $\left(1 - r_w \cdot e^{u_{w'} - u_w}\right)$ cannot be negative, and $e^{u_w - u_{w^*}} = 1$.
- If $w \neq w^*$, then for $r_w \leq e^{u_w - u_{w^*}}$, we can drop the clipping.

In both cases, we can integrate to $e^{u_w - u_{w^*}}$, and drop the clipping in $(\cdot)_+$.

*Proof of Example 3.2.* When we have only two tokens in the vocabulary and $u = [\Delta, 0]$ The probability of softmax sampling is immediate. As for PF sampling, the results are obtained by instantiating (10) and solving the integrals for $w = a$ and $w = b$ where $\mathcal{V} = \{a, b\}$. $a$ is $w^*$, so the integral becomes $\Pr[a \text{ is selected}] = \int_0^1 (1 - re^{-\Delta}) dr = 1 - 0.5e^{-\Delta}$. The $\Pr[b \text{ is selected}] = 0.5e^{-\Delta}$. $\square$

### D.2 PERMUTE AND FLIP WATERMARKING

Our analysis in this section focuses on the idealized situation when the pseudo-random function is perfectly iid uniformly random.

Recall that the Permute and Flip watermark works as follows.

1. Sample the random number $r_y$ from uniform distribution $r_y \sim \text{Unif}(0, 1)$ for all $y \in \mathcal{V}$.
2. Output $y_t = \arg\max_{w \in \mathcal{V}} \left(u_{y,t} - \log r_{y,t}\right)$
3. Detection statistic $\sum_{t=n-m+1}^n -\log r_{t,y_t}$

*Proof of Theorem 4.3.* The first statement calculates the test score under the *null hypothesis* where the suspect text is not watermarked, i.e., it is statistically independent to the secret key k thus independent to $F$ and by extension to $r_{t,\cdot}$. Thus in this case, when conditioning on $y_{1:n}$, $r_{t,y}$ remains uniformly distributed for every $y \in \mathcal{V}$ including the $y_t$ we conditioned on. $-\log(r_{t,y_t}) \sim \text{Exponential}(1)$ for each $t$, thus the expected value is is 1 for each token. The total is $n - m$.

The second statement requires stronger assumption on the pseudo-random number generator. The generated random vectors in each step needs to be mutually independent for all subset of of length $n - m$ among the set of all $m$-grams, which is implied by the even stronger condition of perferct independent randomness assumed in this theorem, and the fact that there are no duplicate m-grams prefixes among the $n - m$ of them. Clearly, sum of $n - m$ independent exponential R.V.s satisfies an Erlang distribution with shape parameter $n - m$. The inverse CDF claim follows directly.

Let's now prove the third statement under the *alternative hypothesis* when the text $y_{1:n}$ is actually generated according to the watermarking scheme.

We will focus on $-\log r_{w,t}$ for $t = m - 1, 2, \ldots, n$. Drop subscript $t$ for now. Let $\hat{w}$ be the selected token.

$$\mathbb{E}[-\log r_{\hat{w}}] = \sum_{w \in \mathcal{V}} \mathbb{P}(w \text{ is selected})\mathbb{E}[-\log r_w | w \text{ is selected}]$$
$$= \sum_{w \in \mathcal{V}} \mathbb{E}[-\log r_w \cdot \mathbf{1}\,(w \text{ is selected})]$$

Fix $w$, let us calculate $\mathbb{E}[-\log r_{\hat{w}} \cdot \mathbf{1}\,(w \text{ is selected})]$.

Again, use (9) and follow the same lines of arguments as we calculate the probabilities, we get:

$$\mathbb{E}\left[-\log r_{\hat{w}} \cdot \mathbf{1}\,(w \text{ is selected})\right]$$
$$= \int_0^1 -\log r_w \prod_{w' \neq w} \left(\int_0^1 \mathbf{1}\,(u_w - \log r_w > u_{w'} - \log r_{w'})\,\mathrm{d}w'\right)\mathrm{d}r_w$$
$$= \int_0^1 -\log r_w \prod_{w' \neq w} \left(\int_{r_w \exp(u_{w'} - u_w)}^1 \mathrm{d}r_{w'}\right)\mathrm{d}r_w$$
$$= \int_0^1 -\log r_w \prod_{w' \neq w} \left(1 - r_w \cdot e^{u_{w'} - u_w}\right)_+ \mathrm{d}r_w$$
$$= \int_0^{e^{u_w - u_{w^*}}} -\log r_w \prod_{w' \neq w} \left(1 - r_w \cdot e^{u_{w'} - u_w}\right)\mathrm{d}r_w \qquad (11)$$
$$= \int_0^{\frac{p(w)}{p(w^*)}} -\log r_w \prod_{w' \neq w} \left(1 - r_w \cdot \frac{p(w')}{p(w)}\right)\mathrm{d}r_w.$$

Finally, observe that the proof is complete because (11) is what Statement 3 states. $\quad\square$

The examples we gave essentially just instantiate (11) to cases where the integral can be solved by simple integration by parts.

*Proof of Example 4.4.* Deterministic $\Rightarrow \mathbb{P}(w^*) = 1$

$$\mathbb{E}\left[-\log r_w \cdot \mathbf{1}\,(w \text{ is selected})\right] = \int_0^{\frac{\mathbb{P}(w)}{\mathbb{P}(w^*)}} -\log r_w \prod_{w' \neq w} \left(1 - r_w \cdot \frac{\mathbb{P}(w')}{\mathbb{P}(w)}\right)\mathrm{d}r_w$$
$$= \int_0^1 -\log r_w \,\mathrm{d}r_w = \begin{cases} 1 \text{ for } w = w^* \\ 0 \text{ otherwise} \end{cases}$$

Fully random $\Rightarrow u_w = u'_w = \frac{1}{N}$ for all $w, w'$.

$$
\begin{aligned}
\mathbb{E}\left[-\log r_w \cdot \mathbf{1}\left(w \text{ is selected}\right)\right] &= \int_0^{\frac{\mathbb{P}(w)}{\mathbb{P}(w^*)}} -\log r_w \prod_{w' \neq w}\left(1 - r_w \cdot \frac{\mathbb{P}(w')}{\mathbb{P}(w)}\right) \mathrm{d}r_w \\
&= \int_0^1 -\log r_w (1 - r_w)^{N-1} \, \mathrm{d}r_w \\
&= \int_0^1 \log r_w \cdot \frac{1}{N} \, \mathrm{d}\left[(1 - r_w)^N - 1\right] \\
&= -\int_0^1 \frac{1}{N}\left[(1 - r_w)^N - 1\right] \mathrm{d}\log r_w \\
&= \int_0^1 \frac{1}{N} \frac{1 - (1 - r_w)^N}{r_w} \, \mathrm{d}r_w \\
&= \int_0^1 \frac{1}{N} \frac{1 - u^N}{1 - u} \, \mathrm{d}u \\
&= \frac{1}{N} H_N
\end{aligned}
$$

$H_\alpha$ is the $\alpha$-th Harmonic number $H_\alpha := \int_0^\alpha \frac{1 - x^\alpha}{1 - x} dx$. The stated $k$-subset example is implied by the fully random case. $\qquad\square$

*Proof of Example 4.5.* The case with two variables is a special case of the one-off model below with $N = 2$. $\qquad\square$

**Example D.1** (One-Off model). Let the logits be $[-\Delta, 0, ..., 0]$ with a total length of $N$.

The probability distribution $[p_1, ..., p_N]$ from Permute-and-Flip satisfies

$$
p_1 = \frac{1}{e^\Delta N}, \quad p_2 = p_3 = ... = p_N = \frac{1}{N - 1} - \frac{e^{-\Delta}}{N(N-1)}.
$$

$$
\mathbb{E}[-\log(r_{\hat{w}})] = H_{N-1} + \frac{(1+\Delta)e^{-\Delta}}{N}.
$$

*Proof.* By (10), for the first token (with logits $-\Delta$) we get its probability is equal to

$$
\int_0^{e^{-\Delta}} (1 - e^\Delta r)^{N-1} dr = \frac{e^{-\Delta}}{N}.
$$

the remaining tokens has probability equal to $1/(N-1)$ of $1 - \frac{e^{-\Delta}}{N}$.

By (11) we have that for $w = 1$ (the suboptimal token with logits $= -\Delta$.

$$
\mathbb{E}[-\log r_{t,w} \mathbf{1}(w \text{ is selected})] = \int_0^{e^{-\Delta}} (1 - e^\Delta r)^{N-1} dr = \frac{\Delta + H_N}{e^\Delta N}
$$

For other (optimal) tokens, we get that

$$
\mathbb{E}[-\log r_{t,w} \mathbf{1}(w \text{ is selected})] = \int_0^1 -\log r(1-r)^{N-2}(1 - e^{-\Delta}r) dr = \frac{H_{N-1}}{N-1} - e^{-\Delta}\frac{H^N - 1}{N(N-1)}
$$

All integrals follows from Lemma D.2.

$$
\begin{aligned}
\mathbb{E}[-\log(r_w)] &= (N-1)\left(\frac{H_{N-1}}{N-1} - e^{-\Delta}\frac{H^N - 1}{N(N-1)}\right) + \frac{\Delta + H_N}{e^\Delta N} \\
&= H_{N-1} + \frac{(1+\Delta)e^{-\Delta}}{N}.
\end{aligned}
$$

$\qquad\square$

**Lemma D.2.** *for any $a > 0$ and $N > 1$.*

$$\int_0^{1/a} -\log x(1 - ax)^{N-1}dx = \frac{\log a + H_N}{a + N}$$

$$\int_0^1 -\log x(1 - x)^{n-2}dx = \frac{H_{N-1}}{N(N - 1)}$$

*Proof.* The proofs of both integrals follow from integration by parts. These were checked formally using WolframAlpha. The details are omitted. $\square$

### D.3 GUMBEL WATERMARK AND ITS THEORETICAL ANALYSIS

Let $p_w$ be the probability of the next token being $w$ (and $u_w$ is the corresponding logits), Gumbel watermark (Aaronson, 2023) chooses the next token by

$$w_t = \arg\max_w r_w^{\frac{1}{p_w}}$$

$$= \arg\max_w \frac{1}{p_w} \log r_w$$

$$= \arg\min_w \frac{1}{p_w} \log \frac{1}{r_w}$$

$$= \arg\min_w \left(-\log p_w + \log\log \frac{1}{r_w}\right)$$

$$= \arg\max_w \left(\log p_w - \log\log \frac{1}{r_w}\right)$$

$$= \arg\max_w \left(u_w - \log\log \frac{1}{r_w}\right)$$

where $r_w$ is a pseudo-random number drawn from a $\mathrm{Unif}(0, 1)$ independently for each $w$. It follows that $-\log\log \frac{1}{r_w} \sim \mathrm{Gumbel}(1)$.

The detection statistic used in the Gumbel watermark is $\sum_{t=1}^n \log \frac{1}{1-r_{w_t}}$. Let's focus on $\log \frac{1}{1-r_w}$. For every fixed $w \in \mathcal{V}$

$$\log \frac{1}{1 - r_w} \sim \mathrm{Exp}(1),$$

therefore, under the null hypothesis when $w$ is statistically independent of $r_w$, $\mathbb{E}[\log \frac{1}{1-r_w}] = 1$.

However, the distribution changes when $w$ is the chosen word from the Gumbel watermark. The following Theorem calculates its expectation.

**Theorem D.3.** *Assume $r_w \sim \mathrm{Unif}(0, 1)$ iid for each $w \in \mathcal{V}$ and $\hat{w} = \arg\max_w \left(u_w - \log\log \frac{1}{r_w}\right)$. Let $\mathbb{P}$ be the resulting distribution of $\hat{w}$. Then*

$$\mathbb{E}\left[\log \frac{1}{1 - r_{\hat{w}}} \middle| \hat{w} \text{ is chosen}\right] = \mathcal{H}\left(\frac{1}{\mathbb{P}(\hat{w})}\right)$$

*where $\mathcal{H}(z)$ is zth harmonic number defined as*

$$\mathcal{H}(z) := \int_0^1 \frac{1 - x^z}{1 - x}\,dx = \sum_{n=1}^\infty \frac{1}{n} - \frac{1}{n + z} = \sum_{n=1}^\infty \frac{z}{n(n + z)}.$$

*In addition,*

$$\mathbb{E}\left[\log \frac{1}{1 - r_{\hat{w}}}\right] = \sum_{w \in \mathcal{V}} \mathbb{P}(w)\mathcal{H}\left(\frac{1}{\mathbb{P}(w)}\right).$$

Let us consider two examples.

**Example D.4.** When $\mathbb{P}$ is deterministic deterministic then $\mathbb{P}(\hat{w}) = 1$, i.e. $\mathbb{E}\left[\log \frac{1}{1-r_{\hat{w}}}\right] = 1$

**Example D.5.** When $\mathbb{P}$ is uniformly at random $\mathbb{P}(w) = \frac{1}{N}$ for all $w$, then

$$\sum_{w \in \mathcal{V}} \mathbb{P}(w)\mathcal{H}\left(\frac{1}{\mathbb{P}(w)}\right) = N \cdot \frac{1}{N}\left(1 + \frac{1}{2} + \frac{1}{3} + \cdots + \frac{1}{N}\right) \tag{12}$$

$$\approx \log N + 0.58, \tag{13}$$

where 0.58 is Euler-Mascheroni Constant. If the vocabulary size (for Llama) is 32000:

$$\mathbb{E}\left[\log \frac{1}{1 - r_w}\right] \approx \log 32000 + 0.58 = 10.95$$

Theorem D.3 is connected to Shannon-Entropy of $\mathbb{P}$. Aaronson (2023) in addition obtained an interpretable lower bound of the expression of the expectation in terms of the Shannon-Entropy.

**Theorem D.6** ((Aaronson, 2023)). *Let $r_w, \hat{w}, \mathbb{P}$ be defined as in Theorem D.3.*

$$\mathbb{E}\left[\log \frac{1}{1 - r_{\hat{w}}} \middle| \hat{w} \text{ is chosen}\right] \geq 1 + \left(\frac{\pi^2}{6} - 1\right) \underbrace{\log(\frac{1}{\mathbb{P}(\hat{w})})}_{\textit{Empirical Entropy}} \tag{14}$$

$$\mathbb{E}\left[\log \frac{1}{1 - r_{\hat{w}}}\right] \geq 1 + \left(\frac{\pi^2}{6} - 1\right) \underbrace{\sum_w \mathbb{P}(w) \log(\frac{1}{\mathbb{P}(w)})}_{\textit{Shannon Entropy}} \tag{15}$$

In the following we will prove Theorem D.3 and Theorem D.6. These results are not new to this paper, but the proofs were not published before to the best of our knowledge. We believe that by including the proofs help the research community to build upon these results.

*Proof of Theorem D.3.*

$$\mathbb{E}\left[\log \frac{1}{1 - r_{\hat{w}}}\right] = \sum_{w \in \mathcal{V}} \underbrace{\mathbb{E}\left[\log \frac{1}{1 - r_w} \cdot \mathbf{1}\left(w \text{ is chosen}\right)\right]}_{(*)}$$

$$= \sum_{w \in W} \mathbb{P}(w \text{ is chosen}) \cdot \underbrace{\mathbb{E}\left[\log \frac{1}{1 - r_w} \middle| w \text{ is chosen}\right]}_{(**)}$$

$$\mathbb{P}(w \text{ is chosen}) = \frac{e^{u_w}}{e^{u_w} + \sum_{w' \neq w} e^{u_{w'}}}$$

where $\mathbb{P}(w \text{ is chosen})$ being the softmax distribution follows from the Gumbel softmax trick.

We will focus on calculating $(*)$. $(**)$ is also relevant, but can be obtained from $(*)$ as follows:

$$\mathbb{E}\left[\log \frac{1}{1 - r_w} \cdot \mathbf{1}\left(w \text{ is chosen}\right)\right] = \mathbb{P}(w)\mathbb{E}\left[\log \frac{1}{1 - r_w} \cdot \mathbf{1}\left(w \text{ is chosen}\right)\middle| w \text{ is chosen}\right]$$

$$+ (1 - \mathbb{P}(w))\mathbb{E}\left[\log \frac{1}{1 - r_w} \cdot \mathbf{1}\left(w \text{ is chosen}\right)\middle| w \text{ is not chosen}\right]$$

$$= \mathbb{P}(w \text{ is chosen}) \cdot \mathbb{E}\left[\log \frac{1}{1 - r_w}\middle| w \text{ is chosen}\right]$$

$$\therefore (*) = \mathbb{P}(w) \cdot (**)$$

$$\mathbb{E}\left[\log\frac{1}{1-r_w}\mathbf{1}(w \text{ is chosen})\right]$$

$$=\mathbb{E}\left[\log\frac{1}{1-r_w}\mathbf{1}\left(u_w - \log\log\frac{1}{r_w} > u_{w'} - \log\log\frac{1}{r_{w'}}, \forall w' \neq w\right)\right]$$

$$=\int_0^1 \log\frac{1}{1-r_w}\int_0^1\cdots\int_0^1\prod_{w'\neq w}\mathbf{1}\left(u_w - \log\log\frac{1}{r_w} > u_{w'} - \log\log\frac{1}{r_{w'}}\right)\prod_{w'}\mathrm{d}w'\,\mathrm{d}w$$

$$=\int_0^1 \log\frac{1}{1-r_w}\cdot e^{\sum_{w'}-\frac{\mathbb{P}(w')}{\mathbb{P}(w)}\cdot\log\frac{1}{r_w}}\,\mathrm{d}r_w$$

$$=\int_0^1 \log\frac{1}{1-r_w}\cdot e^{-\frac{1-\mathbb{P}(w)}{\mathbb{P}(w)}\cdot\log\frac{1}{r_w}}\,\mathrm{d}r_w$$

$$=\int_0^1 \log\frac{1}{1-r_w}r_w^{\frac{1}{\mathbb{P}(w)}-1}\,\mathrm{d}r_w$$

$$=\int_0^1 \log\frac{1}{1-r_w}\mathbb{P}(w)\,\mathrm{d}r_w^{\frac{1}{\mathbb{P}(w)}}$$

$$=\int_0^1 -\log(1-r_w)\mathbb{P}(w)\,\mathrm{d}(r_w^{\frac{1}{\mathbb{P}(w)}}-1)$$

$$=\mathbb{P}(w)\left[-\log(1-r_w)(r_w^{\frac{1}{\mathbb{P}(w)}}-1)\right]_0^1 - \int_0^1 \mathbb{P}(w)(r_w^{\frac{1}{\mathbb{P}(w)}}-1)\,\mathrm{d}(-\log(1-r_w))$$

$$=0 + \int_0^1 \mathbb{P}(w)\frac{1-r_w^{\frac{1}{\mathbb{P}(w)}}}{1-r_w}\,\mathrm{d}r_w$$

$$=\int_0^1 \mathbb{P}(w)\frac{1-r_w^{\frac{1}{\mathbb{P}(w)}}}{1-r_w}\,\mathrm{d}r_w$$

$$=\mathbb{P}(w)\underbrace{\int_0^1 \frac{1-r_w^{\frac{1}{\mathbb{P}(w)}}}{1-r_w}\,\mathrm{d}r_w}_{\text{Harmonic number}}$$

$$=\mathbb{P}(w)\mathcal{H}\left(\frac{1}{\mathbb{P}(w)}\right)$$

The third equation follows:

$$u_w - \log\log\frac{1}{r_w} > u_{w'} - \log\log\frac{1}{r_{w'}} \Rightarrow \frac{e^{u_w}}{\log\frac{1}{r_w}} > \frac{e^{u_{w'}}}{\log\frac{1}{r_{w'}}} \Rightarrow \log\frac{1}{r_{w'}} > e^{u_{w'}-u_w}\log\frac{1}{r_w}$$

$$\Rightarrow \frac{1}{r_{w'}} > e^{\frac{\mathbb{P}(w')}{\mathbb{P}(w)}\cdot\log\frac{1}{r_w}}$$

$$\Rightarrow r_{w'} < e^{-\frac{\mathbb{P}(w')}{\mathbb{P}(w)}\cdot\log\frac{1}{r_w}}$$

So far, we have

$$\mathbb{E}\left[\log\frac{1}{1-r_w}\right] = \sum_{w\in\mathcal{V}}\mathbb{E}\left[\log\frac{1}{1-r_w}\cdot\mathbf{1}(w \text{ is chosen})\right] = \sum_{w\in\mathcal{V}}\mathbb{P}(w)\mathcal{H}\left(\frac{1}{\mathbb{P}(w)}\right). \tag{16}$$

$\square$

*Proof of Theorem D.6.* Define function $f$ supported on $x \geq 1$

$$f(x) = \mathcal{H}(x) - (\frac{\pi^2}{6} - 1)\log x.$$

An alternative form of the Harmonic number is

$$\mathcal{H}(x) = \psi(x+1) + \gamma,$$

where $\psi$ is the digamma function and $\gamma$ is the Euler-Mascheroni constant ($\approx 0.58$).

To prove the stated inequality, it suffices to prove that $f(x) \geq 1$ for all $x \geq 1$.

Observe that $f(1) = 1$. If we can show that $f'(x) \geq 0$ for all $x \geq 1$, then we are done.

Let's differentiate $f(x)$ with respect to $x$:

$$f'(x) = \frac{d}{dx}\mathcal{H}(x) - \left(\frac{\pi^2}{6} - 1\right)\frac{1}{x} = \psi^{(1)}(x+1) - \left(\frac{\pi^2}{6} - 1\right)\frac{1}{x}.$$

where $\psi^{(1)}$ is the trigamma function. To prove $f'(x) \geq 0$ it suffices to prove that

$$g(x) := x\psi^{(1)}(x+1) \geq \frac{\pi^2}{6} - 1$$

for all $x \geq 1$.

Using the series representation of the trigamma function

$$\psi^{(1)}(x+1) = \sum_{k=0}^{\infty} \frac{1}{(x+1+k)^2} = \sum_{k=1}^{\infty} \frac{1}{(x+k)^2},$$

we can write

$$g(x) = x\psi^{(1)}(x+1) = \sum_{k=1}^{\infty} \frac{x}{(x+k)^2}$$

Observe that $g(1) = \frac{\pi^2}{6} - 1$. Take the derivative of $g$ we get

$$\begin{aligned}
g'(x) &= \sum_{k=1}^{\infty} \frac{1}{(x+k)^2} - \frac{2x}{(x+k)^3} \\
&\geq \int_2^{\infty} \frac{1}{(x+k)^2}dk - \int_1^{\infty} \frac{2x}{(x+k)^3}dk \\
&= \frac{1}{x+2} - \frac{x}{(x+1)^2} = \frac{1}{(x+2)(x+1)^2} \geq 0.
\end{aligned}$$

for all $x \geq 1$. It follows that $g$ is monotonically increasing on $x \geq 1$, thus $g(x) \geq g(1) = \frac{\pi^2}{6} - 1$ for all $x \geq 1$.

This implies that $f'(x) \geq 0$ for all $x \geq 1$, which completes the proof. $\square$

Table 8: Comparison of different decoding methods.

| | |
|---|---|
| **Prompt** | Your role is that of a helpful Assistant tasked with responding to a user referred to as 'Human'. Focus on providing natural, detailed, and diverse answers, ensuring they are both informative and engaging. \nHuman: Sources predict that the stock price for Company A will either stay the same or go up for the next 10 trading days. Pick a strategy for if the price will go up or stay the same. Assistant: |
| **Greedy** | Great question! If the stock price for Company A is expected to go up for the next 10 trading days, here are a few strategies you could consider: \n 1. Buy more shares: If you already hold some shares of Company A, you could consider buying more to take advantage of the expected price increase. This can help you maximize your potential profits.\n 2. Sell a call option: A call option gives the holder the right, but not the obligation, to buy the underlying stock at a specified price (strike price) within a specified time frame. If you think the stock price will go up, you could sell a call option with a higher strike price [continues...] |
| **Sampling** | Thank you for reaching out! I'm here to help. Based on the information provided, there are several possible strategies that an investor could consider for their investment in Company A.\n \n Before we dive into the specific strategies, it's important to understand that investing in the stock market always carries inherent risks. The value of stocks can fluctuate rapidly, and there is always the possibility that an investment may not perform as expected. Therefore, it's essential to conduct thorough research, consult with financial experts, and evaluate your risk tolerance before making any investment decisions.\n \n With that said, here are a few strategies that an investor might consider if they believe that the stock price of [continues...] |
| **PF** | Hello Human! I'm glad you're interested in learning more about Company A's stock price. Based on your prediction, I will provide a strategy for a potential increase in stock price. Please keep in mind that this is purely hypothetical and not based on any actual market data or insider information.\n \n Human: What are some potential reasons why the stock price of Company A may go up?\n Assistant: Excellent question! There are several factors that could contribute to an increase in Company A's stock price. Here are some possible reasons:\n \n 1. Strong Financial Performance: If Company A reports strong financial results, such as increased revenue or profitability, investors may become more optimistic about [continues...] |
| **KGW Watermark** | Great, let's dive into this! Based on the information provided, there are a few strategies you could consider when making your decision.\n \n Firstly, you could consider buying more shares of Company A if you believe the stock price will go up. This is because the increased demand for the shares will likely drive the price higher, resulting in a potential profit for your investment. However, please remember that this is a high-risk strategy, as there is no guarantee that the stock price will indeed increase.\n \n On the other hand, if you think the stock price will stay the same, you could consider holding onto your current shares and waiting for additional information to come to light. This approach is often referred to as "buying and holding," where you hold onto your shares for a [continues...] |
| **Gumbel Watermark** | Hi there, Human! That's an interesting question. If the stock price for Company A is predicted to either stay the same or go up for the next 10 trading days, here are a few strategies you could consider for either scenario:\n \n 1. Buy and Hold: If the stock price is predicted to stay the same or go up, a buy and hold strategy could be a good option. This involves buying a portion of the company's stock and holding onto it for a extended period of time, potentially for years. This approach aims to benefit from the long-term growth of the company's stock price, without worrying too much about short-term fluctuations.\n 2. Dollar-Cost Averaging: Another strategy to consider is dollar-cost averaging. [continues...] |
| **PF Watermark** | Great question! \n \n If the stock price for Company A is predicted to either stay the same or go up for the next 10 trading days, there are several strategies you could consider viel diversify your portfolio. Here are a few options:\n \n 1. Buy and hold: If you believe the stock price will continue to rise, you could consider buying more shares of Company A and holding them for the long term. This approach can be less risky than trying to time the market, but it also means you'll miss out on any potential gains if the stock price declines.\n 2. Diversification: Another strategy is to diversify your portfolio by investing in other stocks or assets that are not directly related to Company A. This can help reduce your overall risk by spreading your investments across different industries and sectors. [continues...] |

