# OpenReview forum: "Permute-and-Flip: An optimally stable and watermarkable decoder for LLMs"
_ICLR.cc/2025/Conference — ICLR 2025 Poster_

### Official Review · Reviewer_wmcS · 2024-10-27

**Soundness:** 3
**Presentation:** 2
**Contribution:** 3
**Rating:** 6
**Confidence:** 4

**Summary:**

The article introduces a novel decoder algorithm for LLMs. The proposed algorithm is based on the Permute-and-Flip (PF) method for differential privacy protection of data, which transforms the output of a dataset into independent sampling based on assigned probabilities to data elements. The paper also references an equivalent construction of PF, converting data sampling into random adjustments of the probability distribution of data elements, thereby outputting the element with the highest probability. This decoder algorithm can be used for embedding the output text, and the paper presents a watermark embedding algorithm based on the PF decoder. Compared to existing decoder algorithms, the paper demonstrates that the PF algorithm offers better stability.

**Strengths:**

1. A novel LLM decoder algorithm is proposed, and compared to existing algorithms, the PF decoder offers better randomness, which may provide improved output adaptability for certain application scenarios.

2. The paper proposes a method for embedding watermarks using the PF decoder. It converts candidate output tokens into different values using a pseudo-random function, thereby altering the output probabilities of the tokens to embed the watermark information.

**Weaknesses:**

1. The description of the scheme in the paper is not very straightforward. The PF decoder designed in the paper is not directly based on the original Permute-and-Flip method, but on a variant algorithm. Although it cites (Ding et al., 2021) to illustrate that the two methods are equivalent, this paper is still in a preprint status.

2. The paper does not sufficiently explain the characteristics of the PF decoder. In Theorem 3.1, the paper cites several quantitative calculations from (McKenna & Sheldon 2020) but does not explain their significance. Especially since the original paper was not intended for LLM decoders, the impact on the characteristics of the decoder in this context is unclear.

3. The experimental results show that PF does not demonstrate a significant advantage over existing algorithms. For instance, in text generation, its PPL is not superior to the greedy algorithm, and in watermark embedding detection, its TPR results are also not better than the comparison schemes.

**Questions:**

1. Some symbols in the paper are not fully defined. For example, what does  E_{y∼PF(u)}[u(y)]  mean?

2.In Theorem 3.1,  u^*  is the maximum value of  u(y)  outputs. However, in point 4,  u^* - u(y)  is used. How should one interpret a number being subtracted from a set? Perhaps this is related to the previous question?

---

> ### Author Response · Authors · 2024-11-23
>
> Thank you for your detailed comments. We address each of your concerns below:
>
> > The description of the scheme in the paper is not very straightforward. The PF decoder designed in the paper is not directly based on the original Permute-and-Flip method, but on a variant algorithm. Although it cites (Ding et al., 2021) to illustrate that the two methods are equivalent, this paper is still in a preprint status.
>
> Thank you for pointing this out. We will revise the draft to clarify the derivation and better describe how our PF decoder is related to the original Permute-and-Flip method. The equivalence to the original method, as shown in (Ding et al., 2021), will be more explicitly addressed to ensure the connection is clear.
>
> > The paper does not sufficiently explain the characteristics of the PF decoder. In Theorem 3.1, the paper cites several quantitative calculations from (McKenna & Sheldon 2020) but does not explain their significance. Especially since the original paper was not intended for LLM decoders, the impact on the characteristics of the decoder in this context is unclear.
>
> We appreciate your observation. We have included an explanation tailored to text generation settings in lines 216-227. While the theorem in (McKenna & Sheldon, 2020) was originally developed in the context of differential privacy, we have adapted it to fit text generation settings in this work.
>
> > The experimental results show that PF does not demonstrate a significant advantage over existing algorithms. For instance, in text generation, its PPL is not superior to the greedy algorithm, and in watermark embedding detection, its TPR results are also not better than the comparison schemes.
>
> Our primary emphasis is that the PF watermark achieves the optimal balance between the highest detection accuracy and the lowest perplexity. While some algorithms may excel in either perplexity (e.g., the greedy algorithm) or detectability, the PF watermark is located on the Pareto front, offering the optimal trade-off between these objectives. This balance is illustrated in Figure 3(b), where PF watermarking demonstrates its strength for text generation tasks.
>
>
> > Some symbols in the paper are not fully defined. For example, what does E_{y∼PF(u)}[u(y)] mean?
>
> To clarify: $y$ represents a token. $PF(u)$ denotes the distribution of PF sampling using the logits function $u$. $u(y)$ refers to the logit value for token $y$. Thus, $E_{y \sim PF(u)}[u(y)]$ represents the expected logit value of token $y$ when $y$ is sampled from the PF sampling distribution.
>
> > In Theorem 3.1, u^* is the maximum value of u(y) outputs. However, in point 4, u^* - u(y) is used. How should one interpret a number being subtracted from a set? Perhaps this is related to the previous question?
>
> Both $u^*$ and $u(y)$ are logits values, not sets. Here: - $u^*$ represents the maximum logit value across all tokens. - $u(y)$ refers to the logit value of a specific token $y$.
>
> ---
>
> We hope these clarifications address the reviewers' concerns effectively.

---

### Official Review · Reviewer_vtHu · 2024-11-03

**Soundness:** 3
**Presentation:** 3
**Contribution:** 3
**Rating:** 6
**Confidence:** 4

**Summary:**

The paper introduces the PF decoding method for LLMs, which is designed to improve text generation stability while lowering perplexity compared to traditional sampling methods. This method enhances quality-stability balance and integrates a cryptographic watermark for AI-generated text detection, using a pseudo-random technique that makes it secure and resilient to adversarial modifications. PF decoding achieves lower perplexity than softmax sampling while maintaining stability, and the PF watermark allows for high detection accuracy with precise control over false positives.

**Strengths:**

1. The proposed decoding method for LLMs is novel.
2. This paper provides theoretical proofs to support its statements.

**Weaknesses:**

1. Could authors specify the difference between the PF watermark and the Exponential Minimum watermark proposed in [1].

2. Diversity is an important property of LLM sampling. Are there any metrics or experiments to measure the diversity of different decoding methods?

3. PF watermark provides lower perplexity compared to Gumbel-Max watermark. Could authors present the robustness performance among PF watermark and other watermarking methods, like a token replacement, etc?

4. In Table 1, the authors specify the beam search can not watermark, but [2] uses beam search. Moreover, could authors explain the reason why beam search is not stable?

[1]. http://arxiv.org/abs/2307.15593
[2]. http://arxiv.org/abs/2301.10226

**Questions:**

please see above.

---

> ### Author Response · Authors · 2024-11-23
>
> Thank you for your thoughtful feedback. We address your concerns below:
>
> ---
>
> > Could authors specify the difference between the PF watermark and the Exponential Minimum watermark proposed in [1].
>
> **The Exponential Minimum watermark is essentially the Gumbel watermark**. Kuditipudi et al. utilize the Gumbel sampler to construct a distortion-free watermark but employ fixed randomness to enhance robustness. Specifically, instead of using a pseudorandom function applied to $k$-grams, they cycle through a fixed, pre-determined sequence of values, termed the "key." To detect watermarked text, one iterates over all possible key sequences and tests for similarity between the key and the given response. With $L$ key sequences, this brute-force search increases both the detector's computational complexity and the false positive rate by a factor of $L$. However, this approach allows tolerance for edits up to a fixed rate of changes.
>
> The Exponential Minimum watermark / Gumbel watermark can be formalized as follows:
> $
> y_t = \arg \max_{y \in \mathcal{V}} \frac{u_t(y)}{T} + G_t(y),
> $
> whereas Kuditipudi et al. use:
> $
> y_t = \arg \max_i r_i^{1/p_i}.
> $
> The derivation process is:
>
> $$
> y_t = \arg \max_i r_i^{1/p_i} = \arg \max_i \frac{1}{p_i} \log r_i = \arg \min_i \frac{1}{p_i} \log \frac{1}{r_i} = \arg \min_i \left( -\log p_i + \log \log \frac{1}{r_i} \right) = \arg \max_i \left( \log p_i - \log \log \frac{1}{r_i} \right).
> $$
>
> ---
>
> > Diversity is an important property of LLM sampling. Are there any metrics or experiments to measure the diversity of different decoding methods?
>
> In Table 2, we use the Seq-rep-5 metric to measure text diversity. Seq-rep-5 represents the average repetition rate of duplicate 5-grams in a sequence. PF decoding exhibits lower duplicate 5-grams, indicating good diversity.
>
> Additionally, we calculate the Self-BLEU score to evaluate diversity on the C4 dataset using $T=1.0$ with the Llama 2-7B model. The results are:
>
> | Method       | Self-BLEU Score |
> |--------------|-----------------|
> | Greedy       | 0.1107          |
> | Sampling     | 0.0734          |
> | PF           | 0.0828          |
> | KGW WM       | 0.0609          |
> | Gumbel WM    | 0.0728          |
> | PF WM        | 0.0832          |
>
> Self-BLEU computes the BLEU score for each generated sentence by treating other generated sentences as references. Lower self-BLEU scores indicate higher diversity and less repetition. As shown, the Greedy method has the highest self-BLEU score, suggesting the lowest diversity.
>
> ---
>
> > PF watermark provides lower perplexity compared to the Gumbel-Max watermark. Could authors present the robustness performance of PF watermark against other watermarking methods, such as token replacement?
>
> **We have provided the robustness performance against various paraphrase attack techniques, including paraphrasing and text edits, in Table 4**. These experiments simulate realistic scenarios where an adversary might try to remove the watermark. Our results show that the PF watermark achieves comparable detection performance to both Gumbel and KGW watermark methods when using the same long prefix as the pseudorandom function. This indicates that despite lower perplexity, the PF watermark maintains robustness against common attacks.
>
> ---
>
> > In Table 1, the authors specify that beam search cannot watermark, but [2] uses beam search. Could the authors explain why beam search is not stable?
>
> Our statement refers to beam search's inability to support *provable detectability* in watermarking. This is due to several factors:
> (a) Beam search is a deterministic algorithm, lacking inherent entropy necessary for effective watermarking.
> (b) It's non-autoregressive, making prefix-based pseudo-random designs inapplicable.
> (c) It's sensitive to small logit perturbations, leading to error amplification.
>
> For a detailed explanation of beam search's instability, please refer to Appendix B.2: *Comparison of Decoding Methods in Terms of Stability.*
>
> To demonstrate non-stability, we provide the following counterexample:
>
> - Consider a vocabulary of two tokens. Let $u_t = [0, -1]$ and $\tilde{u}_t = [1, 0]$, where $|u_t - \tilde{u}_t| _{ \infty } \leq \delta$ with $\delta = 1$. Greedy($u_t$) outputs the first token with probability 1, while Greedy ($\tilde{u}_t$) outputs the second token with probability 1. This results in an unbounded importance ratio, proving instability.
>
> Since beam search with a constant score function reduces to greedy decoding, the same counterexample applies, confirming its non-stability.
>
> ---
> We believe these clarifications address the reviewers' concerns effectively and strengthen our paper's contributions.

---

> ### Comment · Reviewer_vtHu · 2024-11-29
>
> Thank you for your response. The authors have addressed my concerns. I have a positive attitude to this paper.

---

### Official Review · Reviewer_5yGg · 2024-11-04

**Soundness:** 2
**Presentation:** 2
**Contribution:** 2
**Rating:** 6
**Confidence:** 4

**Summary:**

The paper proposes sampling text from a language model using the Permute and Flip mechanism of McKenna and Sheldon—treating the token log probabilities as the "score" function to implement the mechanism—and proposes a method for watermarking text sampled from this mechanism.

**Strengths:**

The application of the Permute and Flip (PF) mechanism to sampling text from a language model is an interesting idea. The paper supports its main claims (about the effectiveness of the watermark) with both theoretical and empirical evidence.

**Weaknesses:**

Some parts of the overall problem formulation are not clear. For example, the stated goal is to maximize some utility function, in which case—unlike with differential privacy (i.e., the original context in which PF was proposed)—the optimal decoding strategy should be deterministic. The paper acknowledges as much but nonetheless argues in favor of using sampling via the following:

> That’s because there are other considerations besides text quality when selecting LLM decoders. For example,
computational efficiency and latency are hugely important, since each API call to the logits function is costly. The diversity of the generated text is also important, especially for creative tasks.

Several aspects of the above except are unclear. First, in what sense is greedy decoding more computationally expensive than sampling? Also, if diversity is important then why not formalize this as part of the problem statement (e.g., via the utility function)? Making these desiderata precise would help clarify the main problem statement and aid in comparing to prior work. For similar reasons, it's not clear what the motivation is for the stability property (Definition 2.1). The paper cites data poisoning and jailbreaking attacks as motivation, but the precise connection is not made explicit. The paper also mentions a connection between stability and diversity; however, if the only purpose of stability is to be a proxy for diversity, then why not just explicitly consider diversity? In general, being more explicit and rigorous about the problem formulation / desiderata would be helpful for clarity.

**Questions:**

> A more fair comparison, would be to increase the temperature for PF watermark appropriately so we compare detectability when the suboptimality is aligned. This is shown in Figure 2b. In fact we have added a second baseline that apply Gumbel watermark to the induced sampling distribution from PF-decoding (shown as the dotted line). The distribution induced by PF does not have a simple form, but in our special case, it was worked out in Example 3.2.

After increasing the temperature of the PF watermark so that its detectability is on par with the Gumbel watermark, is the utility of PF still better than that of Gumbel?

---

> ### Author Response · Authors · 2024-11-23
>
> Thank you for your thoughtful feedback. We would like to address your concerns as follows:
>
> > First, in what sense is greedy decoding more computationally expensive than sampling?
>
> We apologize for the confusion in our wording. To clarify, we did not intend to suggest that greedy decoding is computationally expensive.
>
> Our intention was to highlight that an ideal decoding method should possess both computational efficiency and low latency. We will clarify in the revised manuscript that these are desired properties of a decoding method, rather than a comparison with greedy decoding.
>
> > For similar reasons, it's not clear what the motivation is for the stability property (Definition 2.1). The paper cites data poisoning and jailbreaking attacks as motivation, but the precise connection is not made explicit. The paper also mentions a connection between stability and diversity; however, if the only purpose of stability is to be a proxy for diversity, then why not just explicitly consider diversity?
>
> Thank you for raising this point. Our motivation for introducing the stability property is twofold: to enhance diversity and to improve safety. We expanded our discussion in Appendix B to provide more explicit reasoning and connections between stability, diversity, and safety. Appendix B aims to better contextualize stability as a meaningful property, not merely a proxy for diversity.
>
> > After increasing the temperature of the PF watermark so that its detectability is on par with the Gumbel watermark, is the utility of PF still better than that of Gumbel?
>
> Yes, this is precisely the point illustrated in Figure 2(b). When detectability is fixed (y-axis), the utility of the PF watermark remains superior to that of the Gumbel watermark, as evidenced by the smaller x-axis values for PF.
>
> ---
>
> Thank you again for your valuable comments. We believe these clarifications and additional discussions will address your concerns and improve the manuscript.

---

> > ### Comment · Reviewer_5yGg · 2024-11-26
> >
> > The authors have addressed most of my questions and I have adjusted my score accordingly.

---

> > > ### Author Response · Authors · 2024-11-26
> > >
> > > Thank you once again for your support and constructive suggestions!

---

### Official Review · Reviewer_TLR6 · 2024-11-08

**Soundness:** 4
**Presentation:** 4
**Contribution:** 4
**Rating:** 8
**Confidence:** 2

**Summary:**

In this paper, authors propose Permute-and-Flip (PF) decoding for large language models (LLMs),
With reference to McKenna & Sheldon (2020)’s differentially private selection mechanism.
It enjoys the same perturbation-stability guarantees as softmax sampling and achieves substantially lower perplexity.
They also design PF watermark tailored for PF decoding that enables precise control over false positive rates while retaining high true positive rates, with reference to Aaronson (2023)’s Gumbel watermark.
Theorems and experiments indicate that PF decoding a promising new approach for practical applications of LLMs.

**Strengths:**

This paper successfully combines the results of previous studies such as McKenna & Sheldon (2020) and Aaronson (2023) in the context of LLMs research.
Theoretical guarantees are well organized and experimental performance verification is carried out sufficiently, and those make this study reliable.
The proposed method seems to have practical promise.

**Weaknesses:**

I found no serious weakness.

**Questions:**

I have no questions.

---

> ### Author Response · Authors · 2024-11-23
>
> Thank you for your support! We truly appreciate your recognition and positive feedback!

---

### Official Review · Reviewer_acjD · 2024-11-08

**Soundness:** 3
**Presentation:** 3
**Contribution:** 3
**Rating:** 6
**Confidence:** 3

**Summary:**

In this paper, the authors propose a new decoding method called the Permute-and-Flip (PF) decoder. This decoder maintains stability properties similar to standard sampling decoders while achieving up to 2x better performance in the quality-stability tradeoff than traditional sampling methods. A cryptographic watermarking technique from the specifically designed PF decoder has also been developed.
This work's core idea lies in applying PF sampling, initially developed in the context of differential privacy, to watermarking. They also argued that it is the first work to apply this method to LLM decoding and introduce the PF watermarking technique.

**Strengths:**

This paper introduces a new decoding method for large language models, called Permute-and-Flip (PF) decoding, along with a watermarking technique that allows precise control over false positive rates while maintaining high true positive rates. As a result, the approach balances detection accuracy and low perplexity, making it effective for generating high-quality text while preserving watermark detectability.

**Weaknesses:**

- While the PF watermark optimizes better, it reduces entropy in the distribution, potentially weakening the statistical signals that the watermarking scheme can leverage. This suggests that PF decoding may prioritize lower perplexity at the expense of detectability, raising questions about the balance between watermarking efficacy and text quality.
- PF decoding requires tuning the temperature T parameter to maximize detectability. Incorrect temperature settings could degrade performance, potentially limiting practical applications where robustness across diverse settings is essential.
- PF decoding involves a more complex computational process than softmax sampling, which can lead to higher computational costs, especially when handling large vocabulary sets. This complexity may restrict its applicability in real-time or resource-constrained environments.

**Questions:**

The paper claims that the proposed method allows for arbitrarily low false positive rates and high recall when the generated text has high entropy. However, it does not clarify the specific conditions or thresholds that define "high entropy" in the generated text nor the proportion of generated samples expected to meet this criterion. This is particularly confusing given the statement, "PF watermark is better at optimizing (recall Example 3.2 and Figure 1); thus, naturally, the resulting distribution has less entropy to be exploited by the watermarking scheme." This apparent contradiction raises questions about whether PF performance is generally suboptimal due to lower entropy. A precise explanation of these points would clarify the effectiveness and limitations of the PF watermark across different text generation conditions.

**Details Of Ethics Concerns:**

There is no concern.

---

> ### Author Response · Authors · 2024-11-23
>
> Thank you for your insightful feedback. We address your concerns below:
>
> ## Response to W1
>
> > While the PF watermark optimizes better, it reduces entropy in the distribution, potentially weakening the statistical signals that the watermarking scheme can leverage. This suggests that PF decoding may prioritize lower perplexity at the expense of detectability, raising questions about the balance between watermarking efficacy and text quality.
>
> Thank you for raising this point. There is indeed a tradeoff between detectability and quality. As shown in Figure 2a, the PF watermark does not surpass the Gumbel watermark in detectability at a fixed temperature $T$. This is expected since the PF watermark optimizes more efficiently (Example 3.2, Figure 1), resulting in a lower entropy distribution with weaker watermarking signals.
>
> To fairly compare detectability, we adjusted the PF watermark’s temperature to align suboptimality levels. Figure 2b shows that the PF watermark performs comparably or slightly better than the Gumbel watermark in these cases.
>
> Furthermore, as demonstrated in Figure 3b, the PF watermark achieves the optimal balance between high detection accuracy and low perplexity, confirming its practical efficacy.
>
>
> ## Response to W2
>
> > PF decoding requires tuning the temperature T parameter to maximize detectability. Incorrect temperature settings could degrade performance, potentially limiting practical applications where robustness across diverse settings is essential.
>
> This is a valid observation. However, both PF watermark and Gumbel watermark methods require tuning the temperature $T$ to optimize detectability. Table 4 demonstrates the robustness of the PF watermark. It achieves comparable detection performance to the Gumbel and KGW watermarking methods, when using the same long prefix as the pseudorandom function.
>
> ## Response to W3
> > PF decoding involves a more complex computational process than softmax sampling, which can lead to higher computational costs, especially when handling large vocabulary sets. This complexity may restrict its applicability in real-time or resource-constrained environments.
>
> We respectfully disagree with the assertion that PF decoding is computationally more complex than softmax sampling.
>
> While Algorithm 1 in the original submission presents the conceptual steps of PF decoding, Fact 4.2 (derived from Theorem 5 in [1]) demonstrates that Permute-and-Flip Sampling with parameter $T$ can be equivalently implemented as:
> $
> y_t =  \arg\max_{y\in\mathcal{V}}  \frac{\text{logit}_t(y)}{T} +  E_t(y)
> $
> where $E_t(y) \sim \text{Exponential}(1)$ are independent and identically distributed (i.i.d.) for each $t$ (token position) and $y$ (vocabulary token).
>
> This simplified formulation reveals that PF decoding primarily involves sampling exponential noise, which is computationally efficient and can be performed rapidly, even for large vocabulary sets. Consequently, PF decoding does not impose significant computational overhead, making it suitable for real-time or resource-constrained applications.
>
> ## Response to Q1
> > The paper claims that the proposed method allows for arbitrarily low false positive rates and high recall when the generated text has high entropy. However, it does not clarify the specific conditions or thresholds that define "high entropy" in the generated text nor the proportion of generated samples expected to meet this criterion. This is particularly confusing given the statement, "PF watermark is better at optimizing (recall Example 3.2 and Figure 1); thus, naturally, the resulting distribution has less entropy to be exploited by the watermarking scheme." This apparent contradiction raises questions about whether PF performance is generally suboptimal due to lower entropy. A precise explanation of these points would clarify the effectiveness and limitations of the PF watermark across different text generation conditions.
>
> From a text quality perspective, low entropy indicates greedier sampling, which corresponds to better perplexity. If we use perplexity (PPL) as a measure of text quality, lower entropy implies higher quality. However, lower entropy can harm detectability, which explains why the PF watermark does not surpass the Gumbel watermark in detectability at the same temperature. This tradeoff between detectability and quality is central to our analysis, as discussed in lines 409–419. Our experiments also highlight this balance: the PF watermark achieves the best tradeoff between the highest detection accuracy and the lowest perplexity compared to baseline methods.
>
> ---
>
> We believe these revisions address the reviewer's concerns and strengthen the presentation of our work. We are grateful for the opportunity to clarify these points and improve our manuscript.
>
> ---
>
> [1] The Permute-and-Flip Mechanism is Identical to Report-Noisy-Max with Exponential Noise, 2021

---

### Author Response · Authors · 2024-11-25

Dear Reviewers,

We sincerely appreciate your valuable feedback and we have carefully considered and addressed your comments. As the discussion period concludes in two days, we kindly invite you to review our responses. We hope they effectively address your concerns.

Thank you once again for your thoughtful input!

Best regards,

The Authors

---

### Meta-Review · Area_Chair_2vW5 · 2024-12-23

**Metareview:**

**Summary:**
This paper proposes the Permute-and-Flip (PF) decoder for large language models (LLMs), which is provably stable (Theorem 3.1-1), and has optimal stability-perplexity tradeoff (Theorem 3.1-5). It also proposes a new watermarking scheme which they call the PF watermark (Section 4), which allows watermarking of LLM-generated text.

**Strengths:**
- The proposal is based on the idea of PF sampling, which was invented in the literature of differential privacy, and that of the Report-Noisy-Max form of PF sampling, bringing them in the context of language model decoding.
- Theoretical justifications of the proposal are given in Theorems 3.1 and 4.3. Although Theorem 3.1 was a contribution in a different paper in a different context, bringing it to the context of LM decoding is novel.

**Weaknesses:**
Some reviewers raised concerns on description of the proposal and relation to relevant methods.

**Reasons:**
Even though some theoretical results are from existing literature on differential privacy, the interdisciplinary nature of this paper can be evaluated highly.

Minor points:
- Page 6, line 291: exponential random variable(s)
- Facts 4.1 and 4.2: I would prefer enclosing the functions of $y$ to be maximized with a pair of parentheses, as in equation (6), in order to clarify the scope of the argmax operator.

**Additional Comments On Reviewer Discussion:**

Some concerns raised by the reviewers, especially on description of the proposal and relation to relevant methods, have been addressed during the discussion between the authors and the reviewers. The ratings of all the reviewers were on the positive side of the aceptance threshold, even after the author response and discussion between the authors and the reviewers.

---

### Decision · Program_Chairs · 2025-01-22

Accept (Poster)